# naturemethods

# neuromaps: structural and functional interpretation of brain maps

Ross D. Markello ●[1,5], Justine Y. Hansen ●[1,5], Zhen-Qi Liu ●[1], Vincent Bazinet ●[1], Golia Shafiei[1], Laura E. Suárez[1], Nadia Blostein ●[2], Jakob Seidlitz ●[3], Sylvain Baillet ●[1], Theodore D. Satterthwaite[3], M. Mallar Chakravarty[2], Armin Raznahan[4] and Bratislav Misic ●[1] ✉

Imaging technologies are increasingly used to generate high-resolution reference maps of brain structure and function. Comparing experimentally generated maps to these reference maps facilitates cross-disciplinary scientific discovery. Although recent data sharing initiatives increase the accessibility of brain maps, data are often shared in disparate coordinate systems, precluding systematic and accurate comparisons. Here we introduce neuromaps, a toolbox for accessing, transforming and analyzing structural and functional brain annotations. We implement functionalities for generating high-quality transformations between four standard coordinate systems. The toolbox includes curated reference maps and biological ontologies of the human brain, such as molecular, microstructural, electrophysiological, developmental and functional ontologies. Robust quantitative assessment of map-to-map similarity is enabled via a suite of spatial autocorrelation-preserving null models. neuromaps combines open-access data with transparent functionality for standardizing and comparing brain maps, providing a systematic workflow for comprehensive structural and functional annotation enrichment analysis of the human brain.

Imaging and recording technologies such as magnetic resonance imaging (MRI), electro- and magnetoencephalography (EEG and MEG), and positron emission tomography (PET) are used to generate high-resolution maps of the human brain. These maps offer insights into the brain's structural and functional architecture, including gray matter morphometry[1,2], myelination[3–6], gene expression[7,8], cytoarchitecture[9], metabolism[10], neurotransmitter receptors and transporters[11–14], laminar differentiation[15], intrinsic dynamics[16–18] and evolutionary expansion[19–22]. Such maps are increasingly shared on open-access repositories such as NeuroVault[23] or BALSA[24], which, collectively, offer a comprehensive multimodal perspective of the central nervous system. However, these data-sharing platforms are restricted to either surface or volumetric data, and do not integrate standardized analytic workflows.

If researchers generate brain maps in their work, such as task functional MRI activations or case–control cortical thickness contrasts, how can they interpret them? Ideally there should be a way to systematically compare and contextualize generated maps with respect to existing structural and functional annotations, using rigorous statistical methods[25]. In adjacent fields, such as bioinformatics, multiple widely used computational methods for functional profiling and pathway enrichment analysis of gene lists already exist[26,27]. A comparable structural and functional enrichment tool for neuroimaging would have to support three specific capabilities: a method for generating high-quality

[1]Montréal Neurological Institute, McGill University, Montréal, Quebec, Canada. [2]Cerebral Imaging Center, Douglas Mental Health University Institute, McGill University, Montréal, Quebec, Canada. [3]Lifespan Informatics and Neuroimaging Center, Perelman School of Medicine, University of Pennsylvania, Philadelphia, PA, USA. [4]Section of Developmental Neurogenomics, National Institute of Mental Health, Bethesda, MD, USA. [5]These authors contributed equally: Ross D. Markello, Justine Y. Hansen. ✉e-mail: bratislav.misic@mcgill.ca

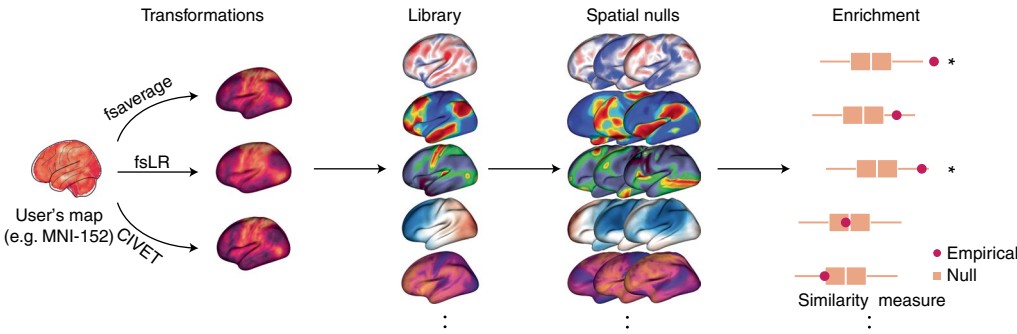

**Fig. 1 | The neuromaps toolbox functionality.** The neuromaps software package features a method for generating high-quality transformations across multiple coordinate systems, a curated repository of brain maps in their native systems, and a method for estimating map-to-map similarity that accounts for spatial autocorrelation.

transformations across multiple coordinate systems, a curated repository of brain maps in their native space, and a method for estimating map-to-map similarity that accounts for spatial autocorrelation.

In the current report we introduce an open-access Python toolbox, neuromaps, to enable researchers to systematically share, transform, and compare brain maps (Fig. 1). First, we generate a set of group-level transformations between four standard coordinate systems that are widely used in neuroimaging and integrate them via a set of accessible, uniform interfaces. Next, we curate more than 40 reference brain maps from the literature that have been published during the past decade to facilitate contextualization of brain annotations. Finally, we implement spatial autocorrelation-preserving null models for statistical comparison between brain maps that will help researchers to perform standardized, reproducible analyses of brain maps. Collectively, this represents a step towards creating systematized knowledge and rapid algorithmic decoding of the multimodal multiscale architecture of the brain.

## Results

The neuromaps software toolbox is available at https://github.com/netneurolab/neuromaps, on PyPi, Zenodo, it exists as a Docker container, and documentation can be found on GitHub pages (https://netneurolab.github.io/neuromaps). In the following section we highlight features available in neuromaps, demonstrate typical workflows enabled by its functionality, and use neuromaps to examine how choice of coordinate system can affect statistical analyses of brain maps.

### The neuromaps data repository

The neuromaps toolbox provides programmatic access to templates for four standard coordinate systems: fsaverage (the default system used by FreeSurfer software, based on 40 normative brains), fsLR (a symmetric version of fsaverage across the left and right hemispheres), CIVET (the default system used by CIVET software) and MNI-152 (developed by the Montreal Neurological Institute using 152 normative MRI scans). For surface-based coordinate systems we distribute template geometry files, sulcal depth maps and average vertex area shape files (computed from Human Connectome Project participants) in standard GIFTI format. For volumetric coordinate systems we distribute T1-, T2-, and proton density-weighted MRI template files, a brain mask, and probabilistic segmentations of gray matter, white matter and cerebrospinal fluid in standard gzipped NIFTI format.

Beyond template files, the neuromaps toolbox offers access to a repository of brain maps obtained from the published literature (Fig. 2 and Supplementary Table 1). These maps were generated using multiple imaging techniques, including MRI, MEG, PET and microarray gene expression. All brain maps except for the genomic gradient are provided in the original coordinate system in which they

were defined to avoid errors caused by successive interpolation. The genomic gradient was upsampled to the fsaverage surface with 10,242 vertices (10k surface) using a $k$-nearest neighbors interpolation. Raw and processed data from the Allen Human Brain Atlas can be accessed at https://abagen.readthedocs.io/en/stable/[28]. Collectively, these maps represent more than a decade of human brain mapping research and encompass phenotypes including the first principal component of gene expression[7], 36 neurotransmitter receptor PET tracer images[14], glucose and oxygen metabolism[10], cerebral blood flow and volume[10], cortical thickness[29], T1-weighted/T2-weighted MRI ratio[30], six canonical MEG frequency bands[29,31], intrinsic timescale[29,31], evolutionary expansion[19], three maps of developmental expansion[19,22], the first 10 gradients of functional connectivity[32], intersubject variability[33] and the first principal component of NeuroSynth-derived cognitive activation[34]. This data repository is organized by tags and can be downloaded directly from neuromaps.

The neuromaps toolbox enables the contextualization of brain maps to a range of molecular, structural, temporal and functional features. This will facilitate an expansion of research questions and enable researchers to bridge brain topographies across several spatial scales and across disciplines outside of their immediate scope[25]. Importantly, the included brain maps are only the start of the neuromaps data repository. The contribution pipeline lets researchers add vertex- and voxel-level brain maps to the toolbox, pending approval from the maintainers. The data repository will therefore become an increasingly rich resource of structural and functional brain annotations. As the toolbox expands it will become more comprehensive, giving the neuroscience field the power to identify cross-disciplinary associations. Information on contributing brain maps can be found in the online documentation for the software (https://netneurolab.github.io/neuromaps/).

### Transformations between coordinate systems

Despite the multiscale, multimodal collection of brain phenotypes in neuromaps, data cannot be readily compared with one another because they exist in different native coordinate systems. Indeed, a common challenge when relating neuroimaging data to the broader literature is finding a common coordinate space or parcellation in which to conduct the analyses. The neuromaps module provides transformations between four supported coordinates systems as well as a standardized set of functions for their application (Supplementary Table 2). Transformation between volumetric- and surface-based coordinate systems relies on a registration fusion framework (Fig. 3a; ref. [35]), whereas transformations between surface-based coordinate systems use a multimodal surface matching (MSM) framework (Fig. 3b; refs. [36,37]). We leverage tools from the Connectome Workbench to provide functionality for applying transformations between surface systems; however, users do not need to interact directly with these Workbench

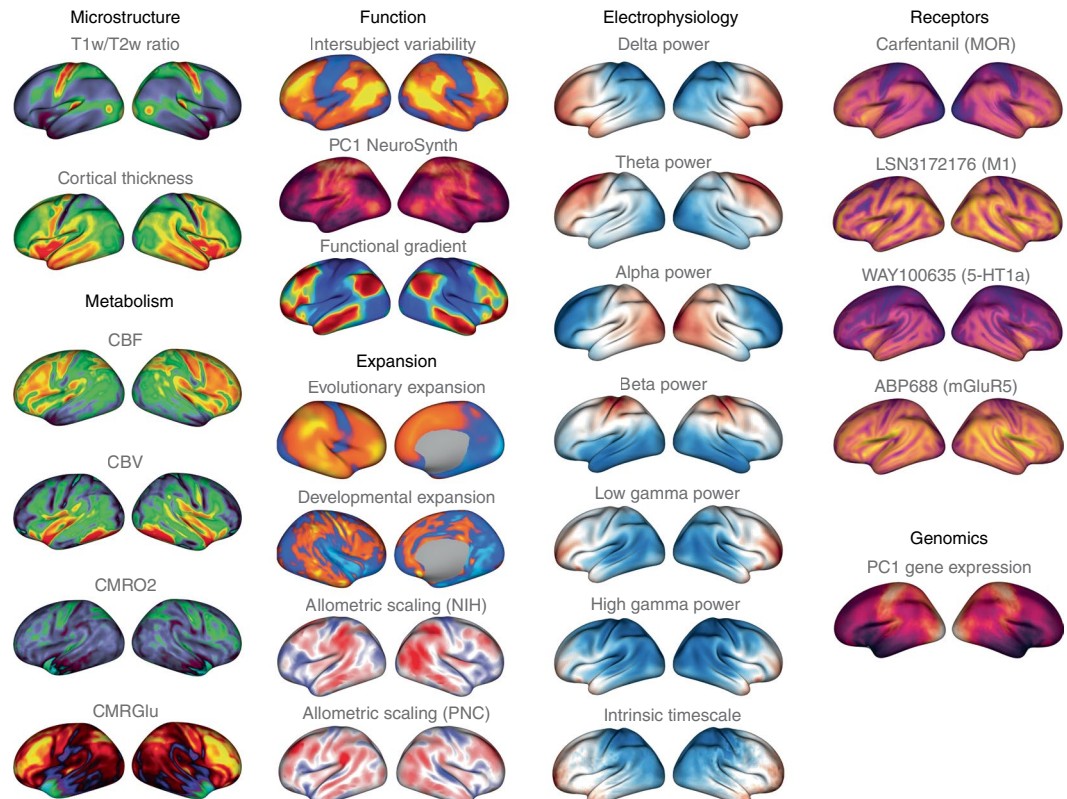

**Fig. 2 | Brain maps from the published literature.** Collection of brain maps obtained from the published literature over the past decade that are currently available in the neuromaps distribution. The maps capture the normative multiscale structural and functional organization of the brain, including molecular, cellular, metabolic and neurophysiological features. Refer to Supplementary Table 1 for more information on the coordinate system, resolution and original publication for each brain map. Colormaps were chosen to maximize similarity with how the data were represented in the original publication. Note that two of the maps (second column: evolutionary and developmental expansion) have data only for the right hemisphere; the intrinsic timescale is log-transformed; a selection of four of the 36 neurotransmitter receptor maps are shown here[14,61–64]; and the genomic gradient is upsampled to the fsaverage 10k surface (for accessing raw and processed Allen Human Brain Atlas data, see https://abagen.readthedocs.io/en/stable/[28]). 5-HT1a, 5-hydroxytryptamine receptor 1A; CBF, cerebral blood flow; CBV, cerebral blood volume; CMRGlu, glucose metabolism; CMRO2, oxygen metabolism; M1, muscarinic receptor 1; mGluR5, metabotropic glutamate receptor 5; MOR, μ-opioid receptor; NIH, National Institutes of Health; PC1, first principal component; PNC, Philadelphia Neurodevelopmental Cohort; T1w/T2w, T1-weighted/T2-weighted MRI.

commands. In addition to transforming individual annotations, the neuromaps software package includes functionalities that receive two brain maps in different spaces as input and return both brain maps in the same space. By default, neuromaps returns the brain maps in the space of the lower-resolution map, which ensures that neuromaps does not artificially create upsampled data. Collectively, the neuromaps toolbox implements robust transformations between coordinate systems to facilitate the standardization of neuroimaging workflows (Fig. 3c,d).

**Spatial null models for comparing brain maps**

The primary goal for transforming maps to a common coordinate system is to statistically compare their spatial topographies. The neuromaps software package uses a flexible framework for examining relationships between brain maps, offering researchers the ability to provide their own image similarity metric or function, and handles any missing data. By default, the primary map comparison workflows use the standard Pearson correlation to test the association between provided maps. The neuromaps comparison workflow also integrates multiple methods of performing spatial permutations for significance testing.

Multiple spatial null model frameworks enable statistical comparison between brain maps while accounting for spatial autocorrelation[4,38–44]; however, the implementation of these models varies and, to date, there has been limited effort to provide a standardized interface for their use. We have incorporated nine null models into the neuromaps toolbox and offer a common user interface for each model that can be integrated with other aspects of the toolbox. Given the computational overhead of these models, our implementations offer mechanisms for caching intermediate results to enable faster re-use across multiple analyses. Spatial null models are enabled by default in the primary map comparison workflows to encourage their broader adoption. Based on prior work that benchmarks the accuracy and computational efficiency of these models[45], we set the non-parametric method as the default for use with surface data[38] and the parameterized generative method as the default for use with volumetric data[43].

**Demonstrating the neuromaps toolbox**

To demonstrate the utility of neuromaps we applied three separate analytic workflows that offer neuroscientific insights. First, we applied the neuromaps toolbox to a volumetric map of cortical thinning derived from comparing T1-weighted MRI scans from $n = 133$ patients with chronic schizophrenia to the T1-weighted MRI scans from $n = 113$ controls from the Northwestern University Schizophrenia Data and Software Tool (NUSDAST) dataset[46] (Fig. 4a). We estimate cortical thinning by applying deformation-based morphometry to T1-weighted MRI scans to calculate the extent of gray matter expansion or contraction in patients relative to controls[47]. We used the neuromaps transformation functions to convert the MNI-152 volumetric schizophrenia brain map ('source map') to the surface space of each of 13 selected brain maps from neuromaps ('target maps'). Next, we correlate the transformed

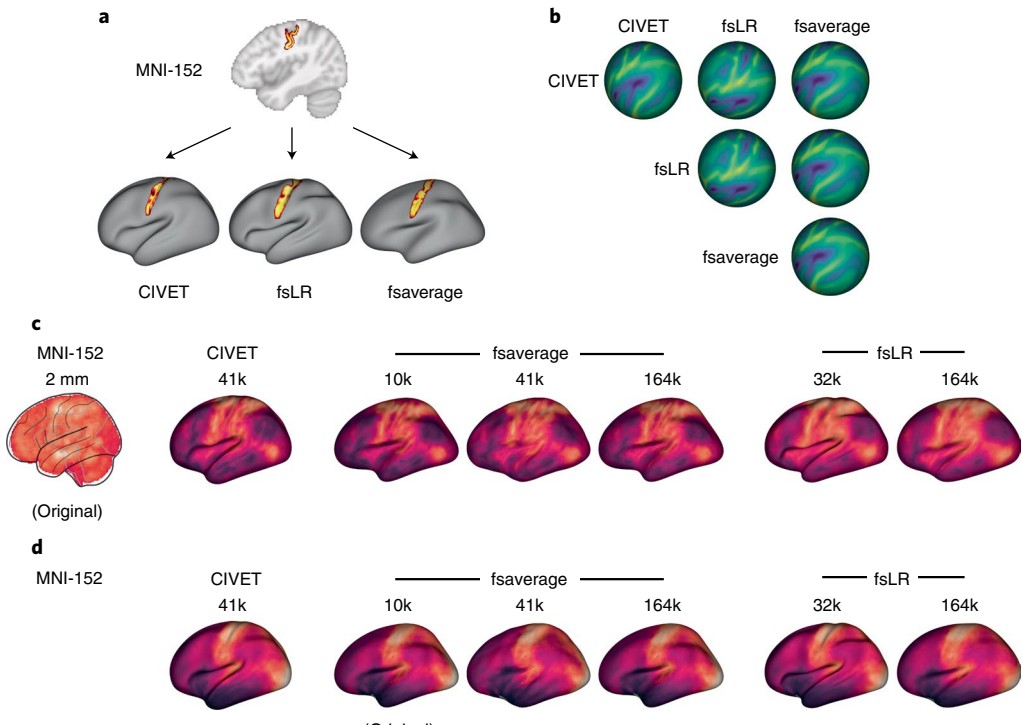

**Fig. 3 | Transformations between coordinate systems. a**, Registration fusion provides a framework for directly projecting group-level volumetric data onto a surface. Here, a probabilistic atlas for the central sulcus[35] has been projected independently onto the CIVET (41k), fsLR (32k) and fsaverage (164k) surfaces. **b**, MSM provides a framework for aligning spherical surface meshes. Here, we show sulcal depth information (originally defined in fsLR space) on spherical meshes that are aligned across the different coordinate systems, where each row represents a different coordinate system and each column represents the space to which that system is aligned. **c**, Example of a volumetric brain map (the first principal component of cognitive terms from NeuroSynth[34]) that has been transformed to all surface-based coordinate systems using alignments derived from registration fusion. **d**, Example of a surface brain map (the first principal component of gene expression from the Allen Human Brain Atlas[7]) that has been transformed to all other surface-based coordinate systems using alignments derived from MSM. Note that because the original data are represented on the cortical surface, transformation to volumetric space is ill-defined and therefore not shown here.

schizophrenia map with each of these 13 target maps (Pearson's $r$) and test the significance using a spatial autocorrelation-preserving null model ('spin test')[38] and false discovery rate correction[48]. We find that schizophrenia-related cortical thinning is enriched in areas with the greatest neurodevelopmental expansion ($r_{NIH} = 0.26$, $P_{spin} = 0.001$; $r_{PNC} = 0.29$, $P_{spin} = 0.001$), consistent with the notion that schizophrenia is a neurodevelopmental disease that affects adolescent development[49,50]. By contrast, we find no evidence to support recent claims that schizophrenia targets specific parts of the unimodal–transmodal processing hierarchy ($r_{functional\ gradient} = -0.08$, $P_{spin} = 0.35$; refs. [51,52]).

Next, we applied the same analytic workflow to a surface-based brain map of evolutionary expansion (already included in neuromaps), which represents cortical surface area expansion from macaque to human[19] (Fig. 4b). Contextualizing this source map with respect to the other 12 selected target maps from the toolbox, we find that areas with the greatest evolutionary expansion have the greatest interindividual variability of regional functional connectivity profiles ($r = 0.58$, $P_{spin} = 0.005$; ref. [33]). This is consistent with the notion that the greatest interindividual differences in brain structure and function are in phylogenetically more recent transmodal cortex[53]. Moreover, we find significant negative correlations with the first principal component of gene expression ($r = -0.51$, $P_{spin} = 0.027$) and intracortical myelin ($r = -0.46$, $P_{spin} = 0.005$), consistent with the idea that phylogenetically newer cortex is characterized by divergence from genomic gradients and canonical stimulus–response circuit configurations[54]. We also find that the map is negatively correlated with cerebral blood volume, which suggests that phylogenetic adaptation is concomitant with greater metabolic efficiency ($r = -0.37$, $P_{spin} = 0.003$). Collectively, these two examples show how the neuromaps toolbox can be used to rigorously generate comprehensive structural and functional annotation enrichment profiles.

Finally, we analyzed a sample of 20 brain maps from the published literature over the past decade (2011–2021), including two microstructural, four metabolic, three functional, four expansion, six band-specific electrophysiological signal power, and one genomic map. We then used neuromaps to transform these maps from their original representation to the space defined by each of four standard coordinate systems, for a total of seven different representations (Fig. 2). Finally, we computed the pairwise correlations between all maps in each of the systems and assessed the statistical significance of these relationships using spatial null models. The goal of this analysis was twofold. First, we sought to assess the extent to which coordinate transforms could influence map-to-map comparisons. Second, given the growing interest in how these system-level maps or 'gradients' are related to one another, we sought to assess patterns of relationships among them[53,55].

For most map-to-map comparisons the choice of coordinate system has a minimal effect: correlation coefficients on average change only by $|r| = 0.018$ (variance of absolute difference, 0.0002; range, −0.10 to 0.079). This is demonstrated by map-to-map correlation matrices (Fig. 5a) and the distribution of correlation changes ($\Delta r$) between pairs of brain annotations in different coordinate spaces (Fig. 5b). Specifically, for a given pair of brain annotations we compute their correlation in the six available surface spaces and resolutions. Next, we plot the distribution of correlation differences between each space and a constant space. Each histogram represents the distribution when a

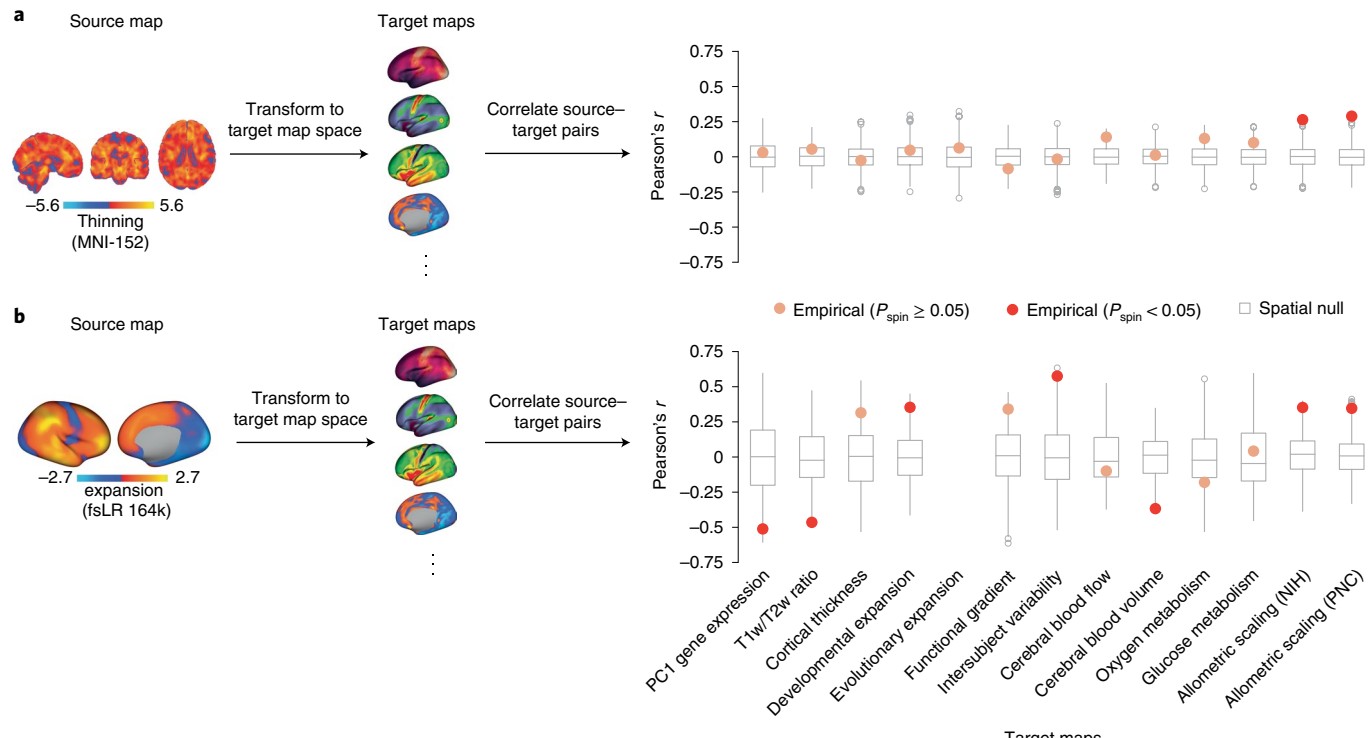

**Fig. 4 | Use of neuromaps to contextualize two exemplar brain maps.**
To demonstrate the utility of neuromaps we use the toolbox to transform, profile and quantitatively assess structural and functional enrichment for two example brain maps (that is, source maps). **a**, A volumetric map of cortical thinning in patients diagnosed with chronic schizophrenia from the NUSDAST repository ($n = 133$ patients versus $n = 113$ controls[46]) was estimated by applying deformation-based morphometry to T1-weighted MRI scans to calculate the extent of gray matter expansion or contraction in patients relative to controls[47]. Warm colors represent regions with greater thinning. The map was transformed to the native space of each brain map to which it was correlated (that is, the target maps). **b**, A surface-based brain map of structural evolutionary expansion represents the ratio of the surface area in humans to

that of macaques, as computed using interspecies surface-based registration[19]. Warm colors indicate regions with greater evolutionary expansion. This map (the source map) was transformed to the native space of each brain map to which it was correlated. Spatial Pearson's correlations were assessed against a two-sided spatial autocorrelation-preserving null model ('spin test')[38]. Points represent the empirical Pearson's correlation between source and target maps (with significance defined as $P_{spin} < 0.05$). In the boxplots the ends of the boxes represent the first (25%) and third (75%) quartiles, the center line (median) represents the second quartile of the null distribution ($n = 1,000$ rotations), the whiskers represent the non-outlier end-points of the distribution, and open circles represent outliers. All correlations were corrected for multiple comparisons[48].

different coordinate space and resolution is defined as the constant space. Although the changes are minimal, there are instances in which associations between maps are statistically significant in one coordinate system and not significant in another. For example, the correlation between the functional gradient and National Institutes of Health (NIH) allometric scaling are significantly correlated in CIVET 41k space ($r = 0.223$, two-sided $P_{spin} = 0.049$) but non-significantly correlated in fsLR 32k space ($r = 0.217$, two-sided $P_{spin} = 0.097$) (Fig. 5c). However, in most cases the $P$ value for these relationships was close to the statistical alpha (that is, $P \leq 0.05$) such that the actual effect size changed only by $r \leq 0.10$. These results are encouraging and suggest that transforming brain annotations between different systems generally preserves their relationships.

Across all examined systems we find that the brain maps tend to form two distinct clusters (Fig. 5d) that largely recapitulate previously established relationships involving anterior–posterior and unimodal–transmodal axes of variation[17,32,56]. One cluster contains maps including the T1-weighted/T2-weighted MRI ratio[57], the principal component of gene expression[4,28], cerebral blood flow and metabolic glucose uptake[10], whereas the other is composed of maps such as the principal functional gradient[32], intersubject functional variability[33] and developmental and evolutionary expansion[19]. This suggests that these brain phenotypes reflect a fundamental organizational principle of the human brain.

## Discussion

Technological and data sharing advances have increasingly moved neuroscience research towards integrative questions rooted in data science. Imaging, recording, tracing and sequencing technology offer an ability to quantify multiple features of neuroanatomy and function with unprecedented detail and depth. Easing the standardization and computation of such comparisons is necessary to ensure that new datasets can be integrated into our broader understanding of the human brain[25]. As the neuromaps toolbox is adopted by the community, annotations from emerging technologies and datasets can be added by users. This will enable maps to be systematically contextualized with respect to multiple canonical annotations from diverse data types and disciplines, resulting in standardized reporting of results, and inspiration for mechanistic follow-up. Ultimately, neuromaps is a step towards integrative analytics for multimodal, multiscale neuroscience.

Given the proliferation of such datasets in recent years, a large body of work has arisen focused on investigating similarities across brain maps[17,18,39,41,56,58,59]. Indeed, researchers have observed substantial concordance in the spatial topology of brain maps derived from a wide variety of phenotypes, suggesting that these maps may reflect a fundamental organizational principle of the human brain. The unimodal–transmodal, or sensory–association, axis is increasingly recognized as a low-dimensional representation of brain organization[6,53,60]. However, large-scale analyses will facilitate comprehensive comparisons across

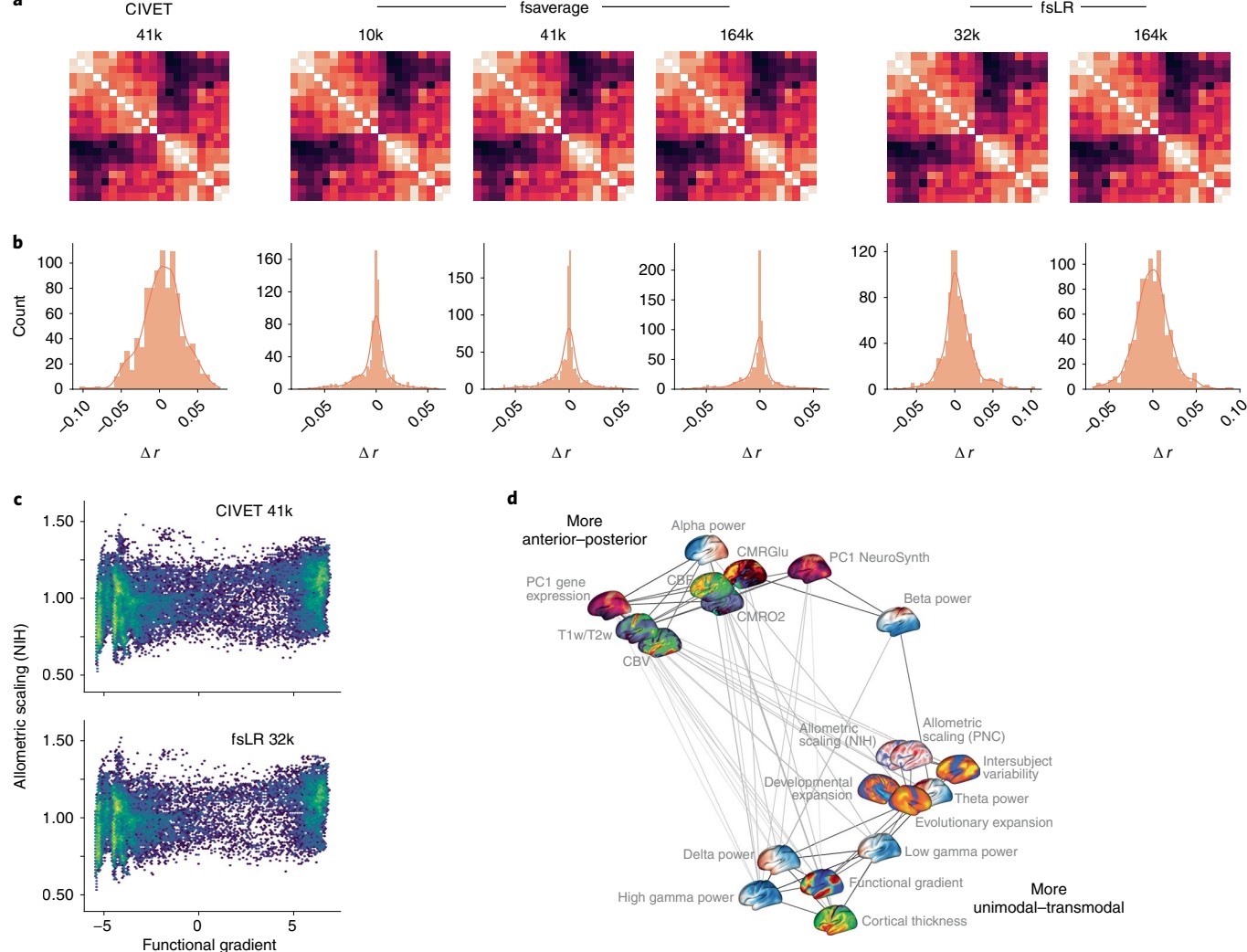

**Fig. 5 | Application of neuromaps to 20 brain maps. a**, Correlation matrices for 20 brain maps in the neuromaps toolbox for each of the surface-based coordinate systems. Because transformations from surface-based to volumetric systems are ill-defined for continuous data we omit those associations. The 20 brain maps are shown in panel **d**. **b**, For each coordinate space and resolution, we show the distribution of correlation changes (Δr) when two brain maps (across all pairs of brain maps) are correlated in the given coordinate space versus all other spaces. **c**, An example of two brain maps (the principal functional gradient from ref. [32] and allometric scaling from ref. [22]), the association between which is significant in one system (CIVET 41k; Pearson's r = 0.223, two-sided $P_{spin}$ = 0.049) and not in another (fsLR 32k; Pearson's r = 0.217, two-sided $P_{spin}$ = 0.097). **d**, A spring-embedded representation of the correlation matrix for the 20 brain maps, shown here for the fsLR 32k system.

multiple brain systems. The question of how gradients representing multiple scales of structural and functional organization interact is an exciting new area of research that can now be addressed using neuromaps.

One consideration that researchers must be aware of when using the neuromaps toolbox is that the provided transformations between coordinate systems are meant to be applied to group-level data; however, in general, when subject-level data are available it is better to reprocess them in the desired coordinate system rather than transforming group-level aggregate data. Unfortunately, in practice, subject-level data for many commonly used brain maps are not available to researchers, and therefore having high-quality transformations between systems is critical to ensuring that analyses are performed in the most accurate manner possible. We have based the provided transformations on state-of-the-art frameworks (that is, registration fusion and multimodal surface matching), which have been rigorously assessed and validated on other datasets[35–37]. Nonetheless, subject-level data that have been registered to a template space can benefit from

the functionalities of neuromaps. For example, subject-level data can be contextualized against the neuromaps library to produce a subject-specific 'fingerprint' of how the individual expresses different structural and functional brain phenotypes. As new frameworks arise for mapping between coordinate systems we will endeavor to provide updated transformations when possible.

Altogether, the current report introduces an open-source Python package, neuromaps, for use in human brain mapping research. As the rate at which new brain maps are generated in the field continues to grow, we hope that neuromaps will provide researchers with a set of standardized workflows to better understand what these data can tell us about the human brain.

## Online content

Any methods, additional references, Nature Research reporting summaries, source data, extended data, supplementary information, acknowledgements, peer review information; details of author contributions and competing interests; and statements of

data and code availability are available at https://doi.org/10.1038/s41592-022-01625-w.

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

## Methods

### Human Connectome Project

Generating transformations between coordinate systems requires high-quality data from a large cohort of individuals; for the transformations in the neuromaps toolbox we use data from the Human Connectome Project (HCP)[29]. Raw T1- and T2-weighted structural MRI data were downloaded for $n = 1,113$ subjects (507 male, age 22–35 years) from the HCP S1200 release, for which informed consent was obtained. After data processing and omission of subjects whose data were not successfully processed using the CIVET pipeline, $n = 1,045$ subjects remained.

**Human Connectome Project processing pipeline.** All structural data were preprocessed using the HCP minimal preprocessing pipelines[29,65]. In brief, T1- and T2-weighted MRI scans were corrected for gradient non-linearity and, when available, images were co-registered and averaged across repeated scans for each individual. Corrected T1-weighted and T2-weighted images were co-registered and cortical surfaces were extracted using FreeSurfer 5.3.0-HCP[2,66]. Subject-level surfaces were aligned to one another using an MSM procedure (MSMAll)[30].

**CIVET processing pipeline.** Images were separately processed with the minc-bpipe-library (https://github.com/CoBrALab/minc-bpipe-library), which performs N4 bias correction, cropping of the neck region, and brain mask generation. Outputs of minc-bpipe-library were then processed through the CIVET pipeline (v2.1.0, ref. [67]), which performs non-linear registration to the MNI International Consortium for Brain Mapping (ICBM) 152 volumetric template, cortical surface extraction, and registration of subject surface meshes to the MNI ICBM 152 surface template. Due to CIVET processing failures the data for $n = 68$ subjects were omitted from further analysis.

### Standard coordinate systems

Here, we describe in brief the four standard coordinate systems (one volumetric and three surface) considered in the current report. Although other coordinate systems are used in neuroimaging research, these four arguably represent the most commonly used systems in the published literature.

**The MNI-152 system.** A significant body of work has been dedicated to explaining what is meant when researchers refer to 'MNI-152 space', given that several variations of this space exist depending on the choice of template[68]. In addition to variations on the MNI-152 template, there exist many other MNI spaces that differ from one another sufficiently to affect downstream analyses[69]. Here, we use the MNI-152 space as defined by the template from the University of Minnesota–Washington University Human Connectome Project group[29], which is a variation of the MNI ICBM 152 non-linear sixth generation symmetric template (identical to the MNI template provided with the FSL distribution[70]). This template was selected because it is the default template in HCP processing pipelines, of which some were used to generate transformations between coordinate systems. This template was created by averaging the T1-weighted MRI scans of 152 healthy young adults that had been linearly and non-linearly (over six iterations) transformed to a symmetric model in Talairach space.

Note, however, that volumetric data are often available in other MNI-152 templates. neuromaps does not currently host functions for transforming between MNI-152 templates (but see https://figshare.com/articles/dataset/MNI_T1_6thGen_NLIN_to_MNI_2009b_NLIN_ANTs_transform/3502238 for transformations between the MNI-152 sixth generation template and the MNI-152 2009b non-linear template). Nonetheless, the transformation functionalities can be applied to data in other MNI-152 templates. These transformations ignore the differences between MNI-152 templates and should therefore be used only when data cannot be registered to the HCP MNI-152 template and if it suits the specific research aim.

**The fsaverage system.** The fsaverage system, used by FreeSurfer, represents data on the fsaverage template, a triangular surface mesh created via the spherical registration of 40 individuals using an energy minimization algorithm to align surface-based features (for example, convexity; refs. [66,71]). In current distributions of FreeSurfer there are five scales of the fsaverage template (fsaverage and fsaverage3–6), ranging in density from 642 to 163,842 vertices per hemisphere. The fsaverage system is roughly aligned to the space of the MNI-305 volumetric system, which was the precursor of the MNI-152 template[72,73].

**The fsLR system.** The fsLR coordinate system was created to overcome perceived shortcomings of the fsaverage system: namely, hemispheric asymmetry[74]. That is, the left and right hemispheres of the fsaverage surface are not in geographic correspondence, such that vertex A in the left hemisphere does not correspond to the same brain region as vertex A in the right hemisphere. The fsLR atlas was created by aligning the two hemispheres of the fsaverage surface into a common hybrid surface, using landmark surface-based registration (originally called the 'fs_LR hybrid atlas'). fsLR templates are available in densities ranging from 32,492 to 163,842 vertices per hemisphere.

**The CIVET system.** The coordinate system used by the CIVET software is a group-averaged surface reconstruction of the individual-participant volumes comprising the volumetric MNI ICBM 152 non-linear sixth generation template[75,76]. In its most commonly used format each hemisphere is represented by 41,962 vertices; a high-resolution version with 163,842 vertices per hemisphere is also available. Because this system is derived from the volumetric MNI template, it ensures that aligned surfaces have good correspondence with volumetric images in the MNI-152 system.

### Generating transformations between systems

Transformation of individual data to a common coordinate system is often performed to account for anatomical differences between individual subjects prior to group aggregation, and makes derived maps more comparable across datasets[72,77]. Data collected from MRI are traditionally represented as volumetric images and are therefore commonly transformed to a standard 'population' image in volumetric space (for example, the MNI ICBM 152 template; refs. [78,79]); however, standardized triangular (that is, 'surface') meshes are increasingly used to represent data as well (for example, the fsaverage, fsLR and CIVET surfaces[66,71,74,76]). Transforming individual, subject-level data between different representations and coordinate systems is non-trivial and has been the focus of substantial research over the past several decades[36,80–87].

Although there are numerous methods for transforming data between coordinate systems, high-quality mappings for group-averaged data are limited[35,88,89]. In creating the neuromaps toolbox, we used two previously validated frameworks to generate transformations between all four standard coordinate systems described above (Fig. 3). All of the transformations were generated using unsmoothed anatomical data and are therefore not biased against data that have been smoothed.

**Registration fusion framework.** Registration fusion is a framework for projecting data between volumetric and surface coordinate systems[35,90]. In its most well-known implementation, researchers used data from the Brain Genomics Superstruct Project[91] to generate non-linear mappings between MNI-152 space and the 164k fsaverage surface[35].

Registration fusion works by generating two sets of mappings for a group of subjects: a mapping between each subject's native image and MNI-152 space, and a mapping between each subject's native image and fsaverage space. These mappings are concatenated (MNI-152 to native

to fsaverage) and then averaged across subjects, yielding a single, high-fidelity mapping that can be applied to new datasets.

Here, we generated mappings via registration fusion between the MNI-152 volumetric and the fsaverage, fsLR and CIVET surface-based coordinate systems using data from the HCP. All mappings used functionality from the Connectome Workbench[92] rather than FreeSurfer to ensure standardization of methodology irrespective of the target coordinate systems.

MNI-152 to CIVET. Unlike for the fsaverage and fsLR surfaces, CIVET surfaces are extracted from subject T1-weighted MRI volumes after the images have been transformed to the standard MNI-152 system. As such, there is no need to generate composite mappings for CIVET surfaces as for the other coordinate systems. Instead, we simply computed the mapping from each subject's MNI-152-transformed T1-weighted MRI volume to the subject's native CIVET surface, and then applied the CIVET-generated surface resampling to register the mapping to the CIVET standard template system. These mappings were then averaged across subjects to generate a single, group-level transformation.

fsaverage, fsLR or CIVET to MNI-152. Although every surface vertex has a corresponding voxel representation in volumetric space, not every voxel has a corresponding vertex representation in surface space. As such, generating transformations from the surface coordinate systems to the MNI-152 volumetric system cannot yield a dense output map. When the current registration fusion framework was proposed[35], a nearest neighbors, ribbon-filling approach was adopted to handle this shortcoming; however, this is a viable approach only when applied to label data (that is, integer-based parcellation images). We reproduce this approach for completeness but caution against the application of surface-to-volume projections for continuous data and omit such projections from our analyses.

**Multimodal surface matching framework.** The MSM framework[36,37] aims to align surfaces defined on different meshes using information from various descriptors of brain structure and function. This procedure has been previously used to generate mappings between the fsaverage and fsLR coordinate systems.

Here, we used MSM to generate a mapping between the CIVET and fsLR systems by aligning HCP subject data processed through the CIVET pipeline with the same data processed through the HCP processing pipeline. Given that MSM requires that input data be provided on spherical surface meshes (a representation not produced in the standard CIVET pipeline) we used FreeSurfer functionality to generate spherical mesh representations and sulcal depth information for each subject's CIVET-derived white matter surfaces. We used these spherical meshes and sulcal depth measurements to drive alignment between the CIVET and fsLR systems via two rounds of the MSM procedure. The first round was used to generate a rotational affine transform to align gross features of the CIVET and fsLR systems; the generated affines were averaged across all subjects and used to seed a second round of finer resolution alignment, similar to the procedure previously described[37]. The final, aligned subject-level spherical surfaces defined in the CIVET system were averaged to create a single, group-level surface that could be used in future transformations.

The CIVET-to-fsaverage mapping was generated as the composite of the transformations between the CIVET-and-fsLR and fsLR-and-fsaverage systems.

**Parcellations.** Performing analyses at the voxel or vertex level can be computationally intensive. The neuromaps software package can be extended to parcellated data and also integrates tools for parcellating volumetric and surface-based data. The base parcellating function assumes that the given parcellation indexes each region with a unique value, where values of 0 are considered background and ignored. Helper functions are provided to flexibly handle alternative parcellation formats. For example, surface parcellations are often

defined in separate left–right GIFTI files for which the same identification numbers are used across both hemispheres, even though each hemisphere has unique parcels. In this case, the user can relabel the parcellation identification numbers such that they are consecutive across hemispheres. This default format was selected to keep surface and volumetric parcellations consistent, and to avoid confusion when hemispheres are not symmetric.

**Published brain maps**
We curated a selection of brain maps from the published literature of the past decade (Fig. 2). Maps were obtained in their original coordinate system, with the exception of the genomic gradient derived from the Allen Human Brain Atlas. The Allen Human Brain Atlas samples across the surface were upsampled to the fsaverage 10k surface using a k-nearest neighbors interpolation before applying principal component analysis. A complete list of maps and their coordinate systems is given in Supplementary Table 1. Some of these maps were originally defined in coordinate systems that are no longer used. In brief, we describe the transformations we used to project these maps to one of the current standard coordinate systems.

**PALS-TA24 to fsLR.** Data obtained from ref. [19] were originally aligned to a study-specific PALS-TA24 template (derived using a similar landmark-based procedure to the PALS-B12 template[93]), which has been supplanted by the fsLR coordinate system. To project data from the PALS-TA24 template to the fsLR system we applied the deformation map provided by the original researchers for transforming data between these spaces using nearest neighbors interpolation.

**CIVET v1 to v2.** The maps obtained from ref. [22] were originally created using surface templates from CIVET v1.1.12; however, with the release of CIVET v2.0.0 in 2014 the population surface templates provided with the CIVET distribution were updated, effectively rendering the older templates redundant. To project data from the CIVET v1.1.12 templates to the CIVET v2.0.0 templates we used a nearest neighbors interpolation, matching vertex coordinates in the newer template to coordinates in the older template and assigning the value corresponding to the closest vertex[22].

**Spatial null frameworks**
Recent research has consistently highlighted the importance of spatially constrained null models when statistically comparing brain maps[38,43,45]. The neuromaps software package integrates nine different spatial null frameworks[45]. These include six spatial permutation models and three parametrized data models, which, collectively, can be constructed for surface-based, volumetric and parcellated data[4,38–44]. Note that four of the null models are adaptations of the original spatial permutation framework[38] when applied to parcellated data[39–42]. These frameworks differ in how they reassign the medial wall (for which most brain maps contain no data), whether that be by discarding missing data[41,42], ignoring the medial wall entirely[40] or reassigning missing data to the nearest parcel[39]. The three parametrized data models circumvent spatial rotations by applying generative frameworks such as a spatial lag model[4], spectral randomization[44] or variogram matching[43].

For analyses in the current report using surface-based coordinate systems, we apply the original spatial permutation framework procedure[38]; for analyses using volumetric systems we apply the variogram-matching method[43]. Null distributions were systematically derived from 1,000 null maps generated by each framework. The mechanism for each null framework used for analyses in the present work is described in brief in the following sections.

**Spatial permutation null model.** The spatial permutation procedure used in the present report generates spatially constrained null distributions by applying random rotations to spherical projections of a cortical

surface[38]. A rotation matrix (**R**) is applied to the three-dimensional coordinates of the cortex (**V**) to generate a set of rotated coordinates (**V**$_{rot}$ = **VR**). The permutation is constructed by replacing the original values at each coordinate with those of the closest rotated coordinate. Rotations are generated independently for one hemisphere and then mirrored across the anterior–posterior axis for the other.

**Variogram estimation null model.** The parametric model used in the present report operates in two main steps: first the values in a given image are randomly permuted, then the permuted values are smoothed and re-scaled to reintroduce spatial autocorrelation characteristic of the original, non-permuted data[43]. Reintroduction of spatial autocorrelation onto the permuted data is achieved via the transformation $y = |\beta|^{1/2}x' + |\alpha|^{1/2}z$, where $x'$ is the permuted data, $z \sim \mathcal{N}(0,1)$ is a vector of random Gaussian noise, and $\alpha$ and $\beta$ are estimated via a least-squares optimization between variograms of the original and permuted data.

## Assessing the impact of coordinate system

When transforming two datasets (that is, a source and target dataset) defined in distinct coordinate spaces to a common system there are at least three options available: transform the source dataset to the system of the target, transform the target dataset to the system of the source, or transform both source and target datasets to an alternate system. If comparisons are being made across several pairs of datasets a fourth option becomes available: always transform the higher resolution dataset to the system of the lower-resolution dataset.

To examine whether the choice of coordinate system affects statistical relationships estimated between brain maps we performed several analyses. First, we transformed a selection of 20 brain maps into every other coordinate system (for example, fsaverage → fsLR, CIVET and MNI-152, fsLR → fsaverage, CIVET and MNI-152, and so on). We then correlated every pair of these brain maps according to each of the four possible resampling options described above. When transforming both source and target datasets to an alternate system (option 3 above), we comprehensively tested every target coordinate system and data resolution. Spatial null models were used to assess the significance of all of the correlations.

### Reporting summary

Further information on research design is available in the Nature Research Reporting Summary linked to this article.

## Data availability

Data used in the present analyses are publicly available on GitHub (https://github.com/netneurolab/neuromaps). The schizophrenia deformation-based morphometry map used in Fig. 4 is derived from the Northwestern University Schizophrenia Data and Software Tool dataset available at https://central.xnat.org/. The Human Connectome Project database is available at https://db.humanconnectome.org/data/projects/HCP_1200.

## Code availability

All code used for data processing, analysis and figure generation directly relies on the following open open-source Python packages: BrainSMASH[43], BrainSpace[44], IPython[94], Jupyter[95], Matplotlib[96], NiBabel[97], Nilearn[98], NumPy[99,100], Pandas[101], PySurfer[102], Scikit-learn[103], SciPy[104], Seaborn[105] and SurfPlot (https://github.com/danjgale/surfplot[106]). Additional software used in the reported analyses includes CIVET (v2.1.1, http://www.bic.mni.mcgill.ca/ServicesSoftware/CIVET[63]), FreeSurfer (v6.0.0, http://surfer.nmr.mgh.harvard.edu/[67]) and the Connectome Workbench (v1.5.0, https://www.humanconnectome.org/software/connectome-workbench[88]). Source code for neuromaps is available on GitHub (https://github.com/netneurolab/neuromaps) and is provided under the Creative Commons Attribution-NonCommercial-ShareAlike 4.0 International License (CC-BY-NC-SA; https://creativecommons.org/licenses/by-nc-sa/4.0/). We have integrated neuromaps with Zenodo, which generates unique digital object identifiers (DOIs) for each new release of the toolbox. Researchers can install neuromaps as a Python package via the PyPi repository (https://pypi.org/project/neuromaps) and can access comprehensive online documentation via GitHub Pages (https://netneurolab.github.io/neuromaps). The neuromaps toolbox is also available as a Docker container (https://hub.docker.com/r/netneurolab/neuromaps/tags), which ensures that the toolbox remains functional even as dependencies are updated and changed. Finally, as an open-source toolbox, neuromaps is open to user suggestions and improvements, ensuring that it remains an evolving resource.

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

## Acknowledgements

R.D.M. acknowledges support from the Fonds de Recherche du Québec – Nature et Technologies and the Canadian Open Neuroscience Platform. J.Y.H. acknowledges support from the Helmholtz International BigBrain Analytics and Learning Laboratory, the Natural Sciences and Engineering Research Council of Canada and the Fonds de Reserche du Québec – Nature et Technologies. S.B. acknowledges support from the National Institutes of Health (NIH) (R01 EB026299), a Discovery Grant from the Natural Science and Engineering Research Council of Canada (436355-13) and the Canadian Institutes of Health Research (CIHR) Canada Research Chair in Neural Dynamics of Brain Systems. T.D.S. acknowledges support from the NIH (R01 MH112847 and R01 MH120482). M.M.C. acknowledges support from the Natural Sciences and Engineering Research Council of Canada, the Canada Research Chairs Program, Healthy Brains for Healthy Lives, and the Fonds du Recherche Québec – Nature et Technologies. B.M. acknowledges support from the Natural Sciences and Engineering Research Council of Canada (NSERC Discovery Grant RGPIN 017-04265), CIHR, the Canada Research Chairs Program, the Healthy Brains for Healthy Lives initiative (HBHL), the Brain Canada Future Leaders Fund and the Michael J. Fox Foundation.

## Author contributions

R.D.M., J.Y.H. and B.M. conceived the study and wrote the manuscript, with valuable revision from all of the authors. R.D.M. developed the software toolbox with help from J.Y.H., Z.-Q.L., V.B., G.S. and L.E.S. J.Y.H. performed the analyses. J.Y.H., G.S., N.B., J.S., S.B., T.D.S., M.M.C. and A.R. contributed data. B.M. was the project administrator.

## Competing interests

R.D.M. is currently employed by Octave Bioscience. The work in this study was performed as part of his graduate studies at McGill University and is in no way related to his employment at Octave Bioscience. All other authors have no competing interests.

## Additional information

**Correspondence and requests for materials** should be addressed to Bratislav Misic.

# nature research

# Reporting Summary

Nature Research wishes to improve the reproducibility of the work that we publish. This form provides structure for consistency and transparency in reporting. For further information on Nature Research policies, see our Editorial Policies and the Editorial Policy Checklist.

## Statistics

For all statistical analyses, confirm that the following items are present in the figure legend, table legend, main text, or Methods section.

| n/a | Confirmed | |
|---|---|---|
| ☒ | ☐ | The exact sample size (*n*) for each experimental group/condition, given as a discrete number and unit of measurement |
| ☒ | ☐ | A statement on whether measurements were taken from distinct samples or whether the same sample was measured repeatedly |
| ☐ | ☒ | The statistical test(s) used AND whether they are one- or two-sided<br>*Only common tests should be described solely by name; describe more complex techniques in the Methods section.* |
| ☒ | ☐ | A description of all covariates tested |
| ☐ | ☒ | A description of any assumptions or corrections, such as tests of normality and adjustment for multiple comparisons |
| ☐ | ☒ | A full description of the statistical parameters including central tendency (e.g. means) or other basic estimates (e.g. regression coefficient) AND variation (e.g. standard deviation) or associated estimates of uncertainty (e.g. confidence intervals) |
| ☐ | ☒ | For null hypothesis testing, the test statistic (e.g. *F*, *t*, *r*) with confidence intervals, effect sizes, degrees of freedom and *P* value noted<br>*Give P values as exact values whenever suitable.* |
| ☒ | ☐ | For Bayesian analysis, information on the choice of priors and Markov chain Monte Carlo settings |
| ☒ | ☐ | For hierarchical and complex designs, identification of the appropriate level for tests and full reporting of outcomes |
| ☐ | ☒ | Estimates of effect sizes (e.g. Cohen's *d*, Pearson's *r*), indicating how they were calculated |

*Our web collection on statistics for biologists contains articles on many of the points above.*

## Software and code

Policy information about availability of computer code

| | |
|---|---|
| Data collection | Data collection software includes the Connectome Workbench (v1.5.0, https://www.humanconnectome.org/software/connectome-workbench). |
| Data analysis | All code used for data processing, analysis, and figure generation directly relies on the following open-source Python packages: BrainSMASH (v0.11.0), BrainSpace (v0.1.2), IPython, Jupyter, Matplotlib (v3.5.0), NiBabel (v3.2.1), Nilearn (v0.8.1), NumPy (v1.21.5), Pandas (v1.3.3), PySurfer, Scikit-learn (v1.1.1), SciPy (v1.6.2), Seaborn (v0.11.2), and SurfPlot (v0.1.0; https://github.com/danjgale/surfplot). Additional software used in the reported analyses includes CIVET (v1.1.12 and v2.1.1, http://www.bic.mni.mcgill.ca/ServicesSoftware/CIVET), FreeSurfer (v6.0.0 and v5.3.0, http://surfer.nmr.mgh.harvard.edu/), and the Connectome Workbench (v1.5.0, https://www.humanconnectome.org/software/connectome-workbench, which is used in HCP's minimal preprocessing pipeline). |

For manuscripts utilizing custom algorithms or software that are central to the research but not yet described in published literature, software must be made available to editors and reviewers. We strongly encourage code deposition in a community repository (e.g. GitHub). See the Nature Research guidelines for submitting code & software for further information.

## Data

Policy information about availability of data

All manuscripts must include a data availability statement. This statement should provide the following information, where applicable:

- Accession codes, unique identifiers, or web links for publicly available datasets
- A list of figures that have associated raw data
- A description of any restrictions on data availability

Data used in the present analyses are publicly available on GitHub (https://github.com/netneurolab/neuromaps).The schizophrenia deformation-based

# Field-specific reporting

Please select the one below that is the best fit for your research. If you are not sure, read the appropriate sections before making your selection.

☒ Life sciences ☐ Behavioural & social sciences ☐ Ecological, evolutionary & environmental sciences

For a reference copy of the document with all sections, see nature.com/documents/nr-reporting-summary-flat.pdf

# Life sciences study design

All studies must disclose on these points even when the disclosure is negative.

| | |
|---|---|
| Sample size | Transformations in the present software toolbox were generated on a cohort of 1045 HCP subjects. Sample size was not chosen, rather, all subjects in the HCP dataset were used. |
| Data exclusions | HCP subjects that did not successfully complete the CIVET processing pipeline were omitted. |
| Replication | The present manuscript proposes a software toolbox and is not experimental. Nonetheless, results are fully replicable using our open code and data. Analyses in Figure 5 were repeated across different coordinate systems. |
| Randomization | The present manuscript proposes a software toolbox and is not experimental. There were no experimental groups and therefore there was no randomization. Each brain map is averaged across all subjects in the sample. |
| Blinding | The present manuscript proposes a software toolbox and is not experimental. There were no experimenal groups and therefore there was no blinding. Each brain map is averaged across all subjects in the sample. |

# Reporting for specific materials, systems and methods

We require information from authors about some types of materials, experimental systems and methods used in many studies. Here, indicate whether each material, system or method listed is relevant to your study. If you are not sure if a list item applies to your research, read the appropriate section before selecting a response.

## Materials & experimental systems

| n/a | Involved in the study |
|---|---|
| ☒ ☐ | Antibodies |
| ☒ ☐ | Eukaryotic cell lines |
| ☒ ☐ | Palaeontology and archaeology |
| ☒ ☐ | Animals and other organisms |
| ☒ ☐ | Human research participants |
| ☒ ☐ | Clinical data |
| ☒ ☐ | Dual use research of concern |

## Methods

| n/a | Involved in the study |
|---|---|
| ☒ ☐ | ChIP-seq |
| ☒ ☐ | Flow cytometry |
| ☐ ☒ | MRI-based neuroimaging |

# Magnetic resonance imaging

## Experimental design

| | |
|---|---|
| Design type | Structural MRI |
| Design specifications | No trials |
| Behavioral performance measures | No behavioural measures |

## Acquisition

| | |
|---|---|
| Imaging type(s) | Structural MRI |
| Field strength | 3T |
| Sequence & imaging parameters | Structural modalities were acquired on a Siemens Skyra 3T scanner and included a T1-weighted MPRAGE sequence at an isotropic resolution of 0.7mm, and a T2-weighted SPACE at an isotropic resolution of 0.7mm. More details on imaging protocols and procedures are available at http://protocols.humanconnectome.org/HCP/3T/imaging- |

| | protocols.html. |
|---|---|
| Area of acquisition | Whole-brain |

Diffusion MRI  ☐ Used  ☒ Not used

## Preprocessing

| Preprocessing software | Preprocessing was done using FSL 5.0.6, FreeSurfer 5.3.0-HCP, and Connectome Workbench v1.1.1. |
|---|---|
| Normalization | Image processing includes correcting for gradient distortion caused by non-linearities, correcting for bias field distortions, and registering the images to a standard reference space. |
| Normalization template | fs_LR_32k surface mesh |
| Noise and artifact removal | No artifact removal was preformed outside of the preprocessing and normalization. |
| Volume censoring | No volume censoring was performed. |

## Statistical modeling & inference

| Model type and settings | No model was applied |
|---|---|
| Effect(s) tested | No effect was tested |

Specify type of analysis:  ☒ Whole brain  ☐ ROI-based  ☐ Both

| Statistic type for inference (See Eklund et al. 2016) | NA |
|---|---|
| Correction | NA |

## Models & analysis

| n/a | Involved in the study |
|---|---|
| ☒ ☐ | Functional and/or effective connectivity |
| ☒ ☐ | Graph analysis |
| ☒ ☐ | Multivariate modeling or predictive analysis |

