## [Peer Review File · Nature Methods]

Peer Review Information

Journal: *Nature Methods*

Manuscript Title: Neuromaps: structural and functional interpretation of brain maps

Corresponding author name(s): Bratislav Mistic

Reviewer Comments & Decisions:

Decision Letter, initial version:

Dear Dr Mistic,

Your Article, "Neuromaps: structural and functional interpretation of brain maps", has now been seen by three reviewers. As you will see from their comments below, although the reviewers find your work of considerable potential interest, they have raised a number of minor concerns. We are interested in the possibility of publishing your paper in *Nature Methods*, but would like to consider your response to these concerns before we reach a final decision on publication. We therefore invite you to revise your manuscript to address these concerns.

- * include a point-by-point response to the reviewers and to any editorial suggestions
- * please underline/highlight any additions to the text or areas with other significant changes to facilitate review of the revised manuscript
- * address the points listed described below to conform to our open science requirements

* ensure it complies with our general format requirements as set out in our guide to authors at www.nature.com/naturemethods

* resubmit all the necessary files electronically by using the link below to access your home page

[Redacted] This URL links to your confidential home page and associated information about manuscripts you may have submitted, or that you are reviewing for us. If you wish to forward this email to co-authors, please delete the link to your homepage.

We hope to receive your revised paper within 4 weeks. If you cannot send it within this time, please let us know. In this event, we will still be happy to reconsider your paper at a later date so long as nothing similar has been accepted for publication at Nature Methods or published elsewhere.

OPEN SCIENCE REQUIREMENTS

REPORTING SUMMARY AND EDITORIAL POLICY CHECKLISTS

DATA AVAILABILITY

We strongly encourage you to deposit all new data associated with the paper in a persistent repository where they can be freely and enduringly accessed. We recommend submitting the data to discipline-specific and community-recognized repositories; a list of repositories is provided here:

<http://www.nature.com/sdata/policies/repositories>

All novel DNA and RNA sequencing data, protein sequences, genetic polymorphisms, linked genotype and phenotype data, gene expression data, macromolecular structures, and proteomics data must be deposited in a publicly accessible database, and accession codes and associated hyperlinks must be provided in the “Data Availability” section.

Please include a “Data availability” subsection in the Online Methods. This section should inform readers about the availability of the data used to support the conclusions of your study, including accession codes to public repositories, references to source data that may be published alongside the paper, unique identifiers such as URLs to data repository entries, or data set DOIs, and any other statement about data availability. At a minimum, you should include the following statement: “The data that support the findings of this study are available from the corresponding author upon request”, describing which data is available upon request and mentioning any restrictions on availability. If DOIs are provided, please include these in the Reference list (authors, title, publisher (repository name), identifier, year). For more guidance on how to write this section please see: <http://www.nature.com/authors/policies/data/data-availability-statements-data-citations.pdf>

CODE AVAILABILITY

Please include a “Code Availability” subsection in the Online Methods which details how your custom code is made available. Only in rare cases (where code is not central to the main conclusions of the paper) is the statement “available upon request” allowed (and reasons should be specified).

MATERIALS AVAILABILITY

ORCID

Nature Methods is committed to improving transparency in authorship. As part of our efforts in this direction, we are now requesting that all authors identified as ‘corresponding author’ on published papers create and link their Open Researcher and Contributor Identifier (ORCID) with their account on the Manuscript Tracking System (MTS), prior to acceptance. This applies to primary research papers only. ORCID helps the scientific community achieve unambiguous attribution of all scholarly contributions. You can create and link your ORCID from the home page of the MTS by clicking on ‘Modify my Springer Nature account’. For more information please visit www.springernature.com/orcid.

Best regards,
Nina

Nina Vogt, PhD
Senior Editor
Nature Methods

Reviewers' Comments:

Reviewer #1:

Remarks to the Author:

This manuscript introduces "neuromaps": a tool to 1/ transform brain maps across coordinate systems, 2/ contextualize brain maps with respect to a set of curated reference maps and 3/ perform statistical comparisons between brain maps (using autocorrelation-preserving null models).

Overall, this is an excellent manuscript that presents a very useful tool for the community. The implemented methods are based on a strong review of state-of-the-art methods. In addition, the authors provide very convincing use cases in which they illustrate how to use the toolbox to reproduce known relationships as well as uncover new biological relationships in the brain.

As an attempt to bridge the gap between different representations used in neuroscience, I believe that this manuscript will be of interest to a broad readership.

Neuromaps is open source and developed publicly on Github. The codebase has an open source license (CC-BY-NC-SA) and includes a code of conduct. The software requirements are well described in the repository.

I only have a few minor comments:

1/ In a few places the authors mention the expected future extensions of this work (e.g. "Additionally, annotations will be regularly added to the repository, resulting in an increasingly rich toolbox.") but it

was not entirely clear from reading the manuscript what is the plan to ensure the maintenance and sustainability of the tool in the future.

2/ The authors acknowledge the existence of multiple variations of the MNI-152 space and justify their choice of the MNI-152 from the Minn-Wash-U HCP. But could they further expand on what they would recommend to a user who would like to compare a new map that uses another MNI-152 space? E.g. Should this map first be transformed to MNI-152 Minn-Wash-U HCP and how? If this is not feasible, can the map be directly used (implicitly ignoring the difference between the MNI-152 template used and the Minn-Wash-U HCP MNI-152 space)? On a similar note, neuroimaging maps are often computed with different level of smoothing, is this likely to impact the transformations / comparisons implemented in neuromaps?

Reviewer #2:

Remarks to the Author:

The paper describes an exciting and seemingly very useful tool for analyzing maps of brain function across studies in common coordinate spaces. They make use of existing tools but weave them together in an accessible format. It also has a small collection of published brain maps across modalities and the functionality to compare brain maps, with the promise of growing this collection. The authors do a good job of thoroughly describing the methods and utilities of the package, as well as providing justification for design choices. The package has clear installation instructions with working demos that highlight the tool's utilities that I found easy to use. There were some sections of the paper that felt like the required heavy lifting from the reader and could use some reworking and more details which are highlighted below. Overall, this paper and tool stand to make an impact in how neuroimagers aggregate and compare data across studies and collections which is a crucial frontier in neuroimaging.

Methods

Page 9. What is the MNI 305 system?

Page 8. "Subject-level surfaces were aligned using a multimodal surface matching (MSMAll) procedure ." does this refer to images being aligned to each other?

Page 9. "Researchers used landmark surface-based registration to align the two hemispheres of the fsaverage surface with a common hybrid target" - can more details be added about what the hybrid target is.

Page 10. “However, helper functions are provided to flexibly handle alternative parcellation formats, for example, where both hemispheres are indexed with the same range of values”. - this is a little confusing since it’s not clear why each hemisphere should need to be indexed separately. Clarification on why this is the default would be helpful. It also seems like this would be important for functional network parcellations and could be helpful to explicitly say

Results

Page 5 “In addition to transforming individual annotations, the neuromaps software package includes functionalities that resample images to each others’ spaces”. - making the distinction between transforming and resampling would be helpful to follow this section

The demonstrating neuromaps toolbox was a little hard to follow and understand exactly what was done. A few specific questions I had were, in the first example was the original map one map comparing schizophrenia to controls? The author write “neuromaps was used to transform the MNI-152 brain map to the surface space of each target brain map”, what are the target brain maps? In the second example “Contextualizing this map with neuromaps , we find that areas with the greatest evolutionary expansion have the greatest inter-individual variability,” - inter individual variability in what? Is this referring to one of the built in maps in neuomaps or variability in the map being tested? I think overall more clarity and thorough explanation of \the data going in and being compared would be helpful. I think adjusting Figure 4 to have a schematic of the exact steps done in the two examples would be really helpful as well.

The authors highlight in the introduction that they focused on transforming group maps from one space to another. It seems like the neuromaps functionality could be useful at the individual level, particularly when one doesn’t want to average across individuals. The author state “In general, when subject-level data are available it is better to reprocess them in the desired coordinate system rather than transforming group-level aggregate data” It would be helpful to expand on this a bit and know how and if this subject level data can be used with neuromaps.

One suggestion for the code: Since neuromaps requires a set of packages with specified version, it might be nice to have a conda environment file or a container etc. for easily creating an environment neuromaps will work in.

Reviewer #3:

Remarks to the Author:

The manuscript from Makello and colleagues introduces a powerful, easy-to-use new Python package (neuromaps) for integrating heterogenous brain mapping data. These datasets span functional neuroimaging and MEG data, PET data, text-to-brain data from NeuroSynth, structural data, gene expression data, and so on. These data are collected on different systems, from different labs, using different mapping / parcellation schemes, so this is a highly non-trivial, useful suite of tools.

To begin, I had to emphasize that I am *not* a neuroimaging specialist, per se, so I cannot give expert analysis of the methods for transforming between coordinate systems. However I have more expertise in data integration and data science, as well as permutation / Monte Carlo methods.

Further, I must apologize for the longer delay in this review, but I actually installed neuromaps to perform some more detailed testing.

Given all of that, I must commend the authors for creating what will surely be a highly impactful tool. I've been wanting something like this for years, and this is an elegant, easy-to-use instantiation of this idea. I was able to get it running with relative ease, and I was able to perform novel analyses with it.

My lab and I are genuinely considering publishing our results that came about from using this new tool that I just installed for the purposes of conducting this review, so consider that a vote of confidence for the approach.

Overall the methods are easy to follow, the installation and use was similarly easy, the manuscript was well-written, and the statistical approaches are sound (with the caveat, as I mentioned earlier, that I am not an imaging expert).

Of course I have a few comments and suggestions for the authors, which I will itemize below.

* First, I think the authors should emphasize that this data integration approach is a *beginning*, not an end. They discuss this, but I think they underemphasize it. Now that this integration framework exists, it should in theory only become more useful and more powerful as more data and data types are pulled into it.

* The authors also undersell, in my opinion, the strength of the contextualization factor that neuromaps permits: currently the vast majority of labs collect data in a vacuum, focusing on their primary approach. However neuromaps provides a easy framework for placing results in the broader context of what is known about the spatial patterns of effects as relate to cognition and disease (NeuroSynth), receipted densities (PET), gene expression (Allen Institute), etc.

* Figure 2, column 3 is labeled "electrophysiology", but technically this is MEG, correct? So it's more "magnetophysiology".

* For the gene expression data, my understanding is that expression is sampled from points. The authors state that they don't up-sample their data--rather they try to keep in in the native format--but I can't figure out from the methods how these point samples of gene expression were used to generate surface maps.

* For Figure 5b, the embedded network image is interesting, but without labels for each of the maps, it's difficult to understand what the takeaway is.

* For Figure 5c, I get the point that arbitrary p-value thresholds can give misleading results, especially considering the flexibility of analysis methods, but these are *very* small differences. Can the authors quantify the distribution of these differences between coordinate systems, perhaps showing a histogram of all the delta-r values for all pairs?

* Relatedly, the authors state that, "choice of coordinate system has a minimal effect: correlations between maps on average only change by $|r| = 0.018$." However without understanding the range, variance, mean, etc. it's impossible to know if this a "big" or "small" effect.

* There are a few results where no statistical effect sizes are given. Specifically the statement that, "schizophrenia-related cortical thinning is enriched in areas that experience the greatest neurodevelopmental expansion," and, "we find no evidence to support recent claims that schizophrenia targets specific parts of the unimodal- transmodal processing hierarchy."

* There are a lot of required packages for neuromaps, and some of them seem like "nice to have" packages, but I don't know why they're in the requirements.txt. For example, why are IPython and Jupiter *required*? Similarly, Matplotlib and Seaborn (and possibly SurfPlot) are nice to have for plotting, but aren't fundamentally required for any of the actual computational processing or data analysis.

Author Rebuttal to Initial comments

Dear Dr. Vogt and Reviewers,

Thank you for the constructive feedback on our first submission, and for the opportunity to revise the manuscript. Following your comments and suggestions, we have thoroughly revised the manuscript. In this letter, we respond to each of the reviewers' comments in detail. Reviewer comments are in bold font and our responses are in regular font.

Reviewer #1:

Remarks to the Author:

This manuscript introduces "neuromaps": a tool to 1/ transform brain maps across coordinate systems, 2/ contextualize brain maps with respect to a set of curated reference maps and 3/ perform statistical comparisons between brain maps (using autocorrelation-preserving null models).

Overall, this is an excellent manuscript that presents a very useful tool for the community. The implemented methods are based on a strong review of state-of-the-art methods. In addition, the authors provide very convincing use cases in which they illustrate how to use the toolbox to reproduce known relationships as well as uncover new biological relationships in the brain.

As an attempt to bridge the gap between different representations used in neuroscience, I believe that this manuscript will be of interest to a broad readership.

Neuromaps is open source and developed publicly on Github. The codebase has an open source license (CC-BY-NC-SA) and includes a code of conduct. The software requirements are well described in the repository.

I only have a few minor comments:

1/ In a few places the authors mention the expected future extensions of this work (e.g. "Additionally, annotations will be regularly added to the repository, resulting in an increasingly rich toolbox.") but it was not entirely clear from reading the manuscript what is the plan to ensure the maintenance and sustainability of the tool in the future.

We now containerize the neuromaps toolbox such that the toolbox will remain functional even as dependencies are updated (see also our response to Reviewer #2 Comment #8). As neuromaps grows, new containers will be automatically released, allowing users to fetch the environment at any stage of neuromaps' development. The first neuromaps container can be found here:

<https://hub.docker.com/r/netneurolab/neuromaps/tags>.

Additionally, in the few months that neuromaps has been live (since January 6, 2022), it has already witnessed suggestions and improvements from community users (7 issues opened and closed by users, 15 forks, and 44 stars). This is encouraging because it is an indication that neuromaps will be successfully maintained by the community. User suggestions will be approved by the toolbox maintainers, which are composed of the Network Neuroscience lab members (who are also frequent users of the toolbox, and familiar with its functionalities).

This has been clarified in the text (“Methods” section, “The neuromaps toolbox” subsection):

“The neuromaps toolbox is also available as a Docker container (<https://hub.docker.com/r/netneurolab/neuromaps/tags>), which ensures the toolbox remains functional even as dependencies are updated and changed. Finally, as an open-source toolbox, neuromaps is open to user suggestions and improvements, ensuring it remains an evolving resource.”

2/ The authors acknowledge the existence of multiple variations of the MNI-152 space and justify their choice of the MNI-152 from the Minn-Wash-U HCP. But could they further expand on what they would recommend to a user who would like to compare a new map that uses another MNI-152 space? E.g. Should this map first be transformed to MNI-152 Minn-Wash-U HCP and how? If this is not feasible, can the map be directly used (implicitly ignoring the difference between the MNI-152 template used and the Minn-Wash-U HCP MNI-152 space)? On a similar note, neuroimaging maps are often computed with different level of smoothing, is this likely to impact the transformations / comparisons implemented in neuromaps?

The volume-to-surface transformation functions within neuromaps are designed using the MNI-152 space defined by the template from the Minn/Wash-U Human Connectome Project group. However, they can also be applied to brain maps in alternative MNI152 spaces. These transformations ignore the differences between MNI152 templates and are therefore suboptimal, albeit still functional. We therefore generally recommend that the HCP-version of the MNI152 template is used, although using other templates will likely make little difference in the transformations (but note that choice of MNI152 template can make a large difference in fMRI processing pipelines (Li et al., 2021)).

Transformations between MNI152 templates are unfortunately underdeveloped. The fMRIPREP software has made some transformation files available between MNI152 6th gen and MNI152 2009b templates, but not for 2009c or 2009a (https://figshare.com/articles/dataset/MNI_T1_6thGen_NLIN_to_MNI_2009b_NLIN_ANTs_transform/3502238).

We have expanded on these points in the text (“Methods” section, “Standard coordinate systems” subsection, “The MNI-152 system” subsection):

“Here we use the MNI-152 space as defined by the template from the Minn/Wash-U Human Connectome Project group (Van Essen et al., 2013), which is a variation of MNI ICBM 152 non-linear 6th generation symmetric template (identical to the MNI template provided with the FSL distribution). This template was selected because it is the default template in HCP processing pipelines, of which some were used to generate transformations between coordinate systems. This template was created by

averaging the T1w MRI images of 152 healthy young adults that had been linearly and non-linearly (over six iterations) transformed to a symmetric model in Talarach space.

Note however that volumetric data is often available in other MNI-152 templates. Neuromaps does not currently host functions for transforming between MNI-152 templates (but see https://figshare.com/articles/dataset/MNI_T1_6thGen_NLIN_to_MNI_2009b_NLIN_ANTs_transform/3502238 for transformations between the MNI-152 6th generation template and the MNI-152 2009b nonlinear template). Nonetheless, the transformation functionalities can be applied to data in other MNI-152 templates. These transformations ignore the differences between MNI-152 templates and should therefore only be used when data cannot be registered to the HCP MNI-152 template and if it suits the specific research aim.”

Regarding smoothing, since the transformations were generated based on unsmoothed anatomical data, applying them to smoothed data will have no impact. The more smoothed the data is, the less anatomically specific it is, which will be propagated to any surface/volume representation of the data. The transformations themselves are not biased against smooth data. We have clarified this in the text (“Methods” section, “Generating transformations between systems” subsection):

“All transformations were generated using unsmoothed anatomical data and are therefore not biased against data that has been smoothed.”

Li, X., Ai, L., Giavasis, S., Jin, H., Feczko, E., Xu, T., ... & Milham, M. P. (2021). Moving beyond processing and analysis-related variation in neuroscience. BioRxiv.

Reviewer #2:

Remarks to the Author:

The paper describes an exciting and seemingly very useful tool for analyzing maps of brain function across studies in common coordinate spaces. They make use of existing tools but weave them together in an accessible format. It also has a small collection of published brain maps across modalities and the functionality to compare brain maps, with the promise of growing this collection. The authors do a good job of thoroughly describing the methods and utilities of the package, as well as providing justification for design choices. The package has clear installation instructions with working demos that highlight the tool's utilities that I found easy to use. There were some sections of the paper that felt like the required heavy lifting from the reader and could use some reworking and more details which are highlighted below. Overall, this paper and tool stand to make an impact in how neuroimagers aggregate and compare data across studies and collections which is a crucial frontier in neuroimaging.

Methods

1) Page 9. *What is the MNI 305 system?*

The MNI-305 template was the first MNI template, consisting of an average of 305 T1-weighted MRI scans, linearly transformed to Talairach space (Evans et al., 1993, Evans et al., 2012). The MNI-152 effectively replaced the MNI-305 template because of its improved image resolution and because it includes more than just the T1-weighted image (i.e. T2w, PDw).

This has been clarified in the text (“Methods” section, “Standard coordinate systems” subsection, “The fsaverage system” subsubsection):

“The fsaverage system is roughly aligned to the space of the MNI-305 volumetric system, which was the precursor of the MNI-152 template (Evans et al., 1993, Evans et al., 2012).”

Evans, A. C., Collins, D. L., Mills, S. R., Brown, E. D., Kelly, R. L., & Peters, T. M. (1993, October). 3D statistical neuroanatomical models from 305 MRI volumes. In 1993 IEEE conference record nuclear science symposium and medical imaging conference (pp. 1813-1817). IEEE.

Evans, A. C., Janke, A. L., Collins, D. L., & Baillet, S. (2012). Brain templates and atlases. *Neuroimage*, 62(2), 911-922.

2) Page 8. *“Subject-level surfaces were aligned using a multimodal surface matching (MSMAll) procedure.” does this refer to images being aligned to each other?*

Yes - we have clarified this in the text (“Methods” section, “Human Connectome Project” subsection, Paragraph #2):

“Subject-level surfaces were aligned to one another using a multimodal surface matching (MSMAll) procedure (Glasser et al., 2016).”

Glasser, M. F., Coalson, T. S., Robinson, E. C., Hacker, C. D., Harwell, J., Yacoub, E., ... & Van Essen, D. C. (2016). A multi-modal parcellation of human cerebral cortex. *Nature*, 536(7615), 171-178.

3) Page 9. *“Researchers used landmark surface-based registration to align the two hemispheres of the fsaverage surface with a common hybrid target” - can more details be added about what the hybrid target is.*

We have clarified the hybrid fsLR surface in the text (“Methods” section, “Standard coordinate systems” subsection, “The fsLR system” subsubsection):

“Proposed by Van Essen et al., 2012, the fsLR coordinate system was created to overcome perceived shortcomings of the fsaverage system: namely, hemispheric asymmetry. That is, the left and right hemispheres of the fsaverage surface are not in geographic correspondence, such that vertex A in the left hemisphere does not correspond to the same brain region as vertex A in the right hemisphere. The fsLR atlas was created by aligning the two hemispheres of the fsaverage surface into a common “hybrid” surface, using landmark surface-based registration (originally called the “fs_LR hybrid atlas”). fsLR templates are available in densities ranging from 32,492--163,842 vertices per hemisphere.”

4) Page 10. “However, helper functions are provided to flexibly handle alternative parcellation formats, for example, where both hemispheres are indexed with the same range of values”. - this is a little confusing since it’s not clear why each hemisphere should need to be indexed separately. Clarification on why this is the default would be helpful. It also seems like this would be important for functional network parcellations and could be helpful to explicitly say

Brain parcellations might be encoded using disparate indexing systems. For example, the medial wall might be indexed by 0, NaN, or “Medial Wall”. Likewise, brain regions might be indexed by consecutive numbers, such that each brain region has a unique identifier. However, parcellations occasionally use the same ID for both hemispheres, despite unique parcels in each hemisphere. This is common when parcellation files are provided for each hemisphere separately. In the case of surface parcellations, neuromaps assumes a specific format (tuple) of left-right GifTI parcellation files, where each brain region has a unique ID (i.e. values are not repeated across hemispheres). This choice avoids confusion when hemispheres are not symmetric (and identical IDs do not refer to homologous brain regions). It is also more consistent with volumetric parcellations, where each brain region has a unique identifier and hemispheres are not separated. Notably, in the case where regions in a parcellation are bilateral and cover vertices from both left and right hemispheres (as is the case in functional network parcellations (Yeo & Krienen et al., 2011)), the two GifTI parcellation files should be indexed using the same IDs.

We have clarified why this parcellation format is the default in the text (“Methods” section, “Generating transformations between systems” subsection, “Parcellations” subsection):

“Performing analyses at the voxel or vertex level can be computationally intensive. The neuromaps software package can be extended to parcellated data and also integrates easy-to-use tools for parcellating volumetric and surface data. The base parcellating function assumes that the given parcellation indexes each region with a unique value, where values of 0 are considered background and ignored. Helper functions are provided to flexibly handle alternative parcellations formats. For example, surface parcellations are often defined in separate left-right gifTI files where the same ID numbers are used across both hemispheres, even though each hemisphere has unique parcels. In this case, the user can relabel the parcellation IDs such that they are consecutive across hemispheres. This default format

was selected to keep surface and volumetric parcellations consistent, and to avoid confusion when hemispheres are not symmetric.”

Yeo, B. T., Krienen, F. M., Sepulcre, J., Sabuncu, M. R., Lashkari, D., Hollinshead, M., ... & Buckner, R. L. (2011). The organization of the human cerebral cortex estimated by intrinsic functional connectivity. *Journal of neurophysiology*.

Results

5) Page 5 “In addition to transforming individual annotations, the neuromaps software package includes functionalities that resample images to each others’ spaces”. - making the distinction between transforming and resampling would be helpful to follow this section

In this context, “transforming” and “resampling” refer to the same procedure of changing the coordinate space of a brain map. The original text was unclear so we have rephrase this section (“Results” section, “Transformations between coordinate systems” subsection):

“In addition to transforming individual annotations, the neuromaps software package includes functionalities that receive two brain maps in different spaces as input and returns both brain maps in the same space. By default, neuromaps returns the brain maps in the space of the lower-resolution map, which ensures neuromaps does not artificially create new (i.e. it does not upsample) data.”

6) The demonstrating neuromaps toolbox was a little hard to follow and understand exactly what was done. A few specific questions I had were, in the first example was the original map one map comparing schizophrenia to controls? The authors write “neuromaps was used to transform the MNI-152 brain map to the surface space of each target brain map”, what are the target brain maps? In the second example “Contextualizing this map with neuromaps, we find that areas with the greatest evolutionary expansion have the greatest inter-individual variability,” - inter individual variability in what? Is this referring to one of the built in maps in neuromaps or variability in the map being tested? I think overall more clarity and thorough explanation of the data going in and being compared would be helpful. I think adjusting Figure 4 to have a schematic of the exact steps done in the two examples would be really helpful as well.

We thank the Reviewer for bringing this up and apologize for the unclear demonstration of neuromaps. We respond to the Reviewer’s specific questions below:

- Re the schizophrenia map: this volumetric map comes from applying deformation based morphometry to T1w scans from healthy controls and patients with chronic schizophrenia. Specifically, T1w images for 113 healthy controls were averaged together to construct an average “healthy” T1w map, and T1w images for 133 patients with chronic schizophrenia were average to create a “schizophrenia” T1w map.

Then, deformation based morphometry compares the two averaged T1w images and estimates the amount of deformation (shrinkage or expansion) that occurs from the control map to the schizophrenia map. In other words, it is a single volumetric map of schizophrenia-related deformation. This deformation map is what we contextualize using neuromaps, and we refer to it as the “source map”.

- Re target maps: the target maps are each map that the schizophrenia map is being compared with. For example, when the schizophrenia map was correlated to the PC1 gene expression map, it was first transformed to the space of the PC1 gene expression map. This was done because the schizophrenia map was volumetric and therefore would only ever be downsampled to match the surface-based target maps. The same applies to the evolutionary expansion map because all target maps were either in the same space (fsLR 164k) or in a coarser space.

- Re inter-individual variability: this map is derived from Mueller et al., 2013 and represents a map of variability of functional connectivity across individuals. This is a map included in neuromaps.

We have thoroughly reworked this section to clarify what was done (“Results” section, “Demonstrating the neuromaps toolbox”, Paragraph #1-2):

“To demonstrate the utility of neuromaps, we applied three separate analytic workflows which provide novel, and confirm previously established, neuroscientific insights. First, we applied the neuromaps toolbox to a volumetric map of cortical thinning derived from comparing T1w images from N=133 patients with chronic schizophrenia to the T1w images from N=113 controls from the Northwestern University Schizophrenia Data and Software Tool (NUSDAST) dataset (Kogan et al., 2016; Figure 4a). Cortical thinning was estimated by applying deformation based morphometry to T1w scans to calculate the extent of grey matter expansion or contraction in patients relative to controls, which results in a single map of schizophrenia-related cortical thinning (for methodological details see Shafiei et al., 2021). The neuromaps data repository facilitates the contextualization of novel brain maps against various structural and functional annotations. As a demonstration, neuromaps transformation functions were used to convert the MNI-152 volumetric schizophrenia brain map (“source map”) to the surface space of each of 13 selected brain maps from neuromaps (“target maps”). The transformed schizophrenia map was then correlated with each of these 13 target maps (Pearson’s r) and significance was assessed using a spatial autocorrelation-preserving null model (Alexander-Bloch et al., 2018) and false discovery rate correction (Benjamini & Hochberg, 1995). We find that schizophrenia-related cortical thinning is enriched in areas that experience the greatest neurodevelopmental expansion ($r_{NIH}=0.26$, $pspin=0.001$; $r_{PNC}=0.29$, $pspin=0.001$), consistent with the notion that schizophrenia is a neurodevelopmental disease that affects adolescent development (Murray et al., 1987, Rapoport et al., 2012). On the other hand, we find no evidence to support recent claims that schizophrenia targets specific parts of the unimodal-transmodal processing hierarchy ($r_{FC}=-0.08$, $pspin=0.35$; Dong et al., 2021, Dong et al., 2020).

Next, we applied the same analytic workflow to a surface-based brain map of evolutionary expansion (already included in neuromaps), which represents cortical surface area expansion from macaque to human (Hill et al., 2010; Figure 4b).

Contextualizing this source map with respect to the other 12 selected target maps from the toolbox, we find that areas with the greatest evolutionary expansion have the greatest inter-individual variability of regional functional connectivity profiles ($r=0.58$, $p_{spin}=0.005$; Mueller et al., 2013). This is consistent with the notion that the greatest inter-individual differences in brain structure and function are in phylogenetically newer transmodal cortex (Sydnor et al., 2021). Moreover, we find significant anticorrelations with the genomic PC1 ($r=-0.51$, $p_{spin}=0.027$) and intracortical myelin ($r=-0.46$, $p_{spin}=0.005$), consistent with the idea that phylogenetically newer cortex is characterized by divergence from genomic gradients and canonical stimulus-response circuit configurations (Buckner & Krienen, 2013). Interestingly, we also find that the map is anticorrelated with cerebral blood volume - a novel result that suggests that phylogenetic adaptation is concomitant with greater metabolic efficiency ($r=-0.37$, $p_{spin}=0.003$). Collectively, these two examples show how the neuromaps toolbox can be used to rigorously generate comprehensive structural and functional annotation enrichment profiles, yielding novel biological insight.”

We have also updated the figure to show more clearly what is being done:

“Figure 4. Applying neuromaps to contextualize two exemplar brain maps | To demonstrate the utility of neuromaps, we use the toolbox to transform, profile, and quantitatively assess structural and functional

enrichment for two example brain maps (“source maps”). Spatial correlations were assessed against a spatial autocorrelation-preserving null model (“spin test”) (Alexander-Bloch et al., 2018). Points represent the empirical correlation between source and target maps (red dots are significant ($p_{\text{spin}} < 0.05$)). Boxplots represent the 1st, 2nd (median) and 3rd quartiles of the null distribution, whiskers represent the non-outlier end-points of the distribution, and open circles represent outliers. All correlations were corrected for multiple comparisons (Benjamini & Hochberg, 1995). (a) A volumetric map of cortical thinning in patients diagnosed with chronic schizophrenia from the NUSDAST repository (N = 133 patients vs. N = 113 controls; (Kogan et al., 2016)) was estimated by applying deformation based morphometry to T1w scans to calculate the extent of grey matter expansion or contraction in patients relative to controls (for methodological details see Shafiei et al., 2020). Warm colours represent regions with greater thinning. The map was transformed to the native space of each brain map to which it was correlated (“target maps”, named on the x-axis of the boxplot). (b) A surface-based brain map of structural evolutionary expansion represents the human to macaque surface area ratio, as computed using interspecies surface-based registration (Hill et al., 2010). Warm colours indicate regions with greater evolutionary expansion. This map (the “source map”) was transformed to the native space of each brain map to which it was correlated (“target maps”, named on the x-axis of the boxplot).”

7) The authors highlight in the introduction that they focused on transforming group maps from one space to another. It seems like the neuromaps functionality could be useful at the individual level, particularly when one doesn't want to average across individuals. The author state “In general, when subject-level data are available it is better to reprocess them in the desired coordinate system rather than transforming group-level aggregate data”. It would be helpful to expand on this a bit and know how and if this subject level data can be used with neuromaps.

This is an interesting point that we did not explore in the original manuscript. We focus on group-averaged maps (all brain maps included in the neuromaps library are group-averages) but can readily handle subject-level data that exists in one of the four template spaces (MNI-152, fsaverage, fsLR, CIVET).

Individual data in native space is not compatible with neuromaps.

We have added the following paragraph to the revised manuscript to clarify how neuromaps can be used with subject-level data (“Discussion” section, Paragraph #5):

“One consideration researchers must be aware of when using the neuromaps toolbox is that the provided transformations between coordinate systems are meant to be applied to group-level data; however, in general, when subject-level data are available it is better to reprocess them in the desired coordinate system rather than transforming group-level aggregate data. Unfortunately, in practice, subject-level data for many commonly-used brain maps are not available to researchers, and so having high-quality transformations between systems is critical to ensuring that analyses are performed in the

most accurate manner possible. We have based the provided transformations on state-of-the-art frameworks (i.e., registration fusion and multimodal surface matching) which have been rigorously assessed and validated on other datasets (Wu et al., 2018, Robinson et al., 2014, Robinson et al., 2014). Nonetheless, subject-level data that has been registered to a template space can benefit from neuromaps' functionalities. For example, subject-level data can be contextualized against the neuromaps library to produce a subject-specific "fingerprint" of how the individual expresses different structural and functional brain phenotypes. Moving forward, as new frameworks arise for mapping between coordinate systems we will endeavor to provide updated transformations when possible."

8) One suggestion for the code: Since neuromaps requires a set of packages with specified version, it might be nice to have a conda environment file or a container etc. for easily creating an environment neuromaps will work in.

We thank the Reviewer for this suggestion. Neuromaps is now available as a Docker container and an automatic Github workflow has been set up to push new Docker containers to Docker Hub when neuromaps updates are released. The Docker container can be found here: <https://hub.docker.com/r/netneurolab/neuromaps/tags>.

This has been clarified in the text ("Methods" section, "The neuromaps toolbox" subsection):

"The neuromaps toolbox is also available as a Docker container (<https://hub.docker.com/r/netneurolab/neuromaps/tags>), which ensures the toolbox remains functional even as dependencies are updated and changed. Finally, as an open-source toolbox, neuromaps is open to user suggestions and improvements, ensuring it remains an evolving resource."

Reviewer #3:

Remarks to the Author:

The manuscript from Markello and colleagues introduces a powerful, easy-to-use new Python package (neuromaps) for integrating heterogeneous brain mapping data. These datasets span functional neuroimaging and MEG data, PET data, text-to-brain data from NeuroSynth, structural data, gene expression data, and so on. These data are collected on different systems, from different labs, using different mapping / parcellation schemes, so this is a highly non-trivial, useful suite of tools.

*To begin, I had to emphasize that I am *not* a neuroimaging specialist, per se, so I cannot give expert analysis of the methods for transforming between coordinate systems. However I have more expertise in data integration and data science, as well as permutation / Monte Carlo methods.*

Further, I must apologize for the longer delay in this review, but I actually installed neuromaps to perform some more detailed testing.

Given all of that, I must commend the authors for creating what will surely be a highly impactful tool. I've been wanting something like this for years, and this is an elegant, easy-to-use instantiation of this idea. I was able to get it running with relative ease, and I was able to perform novel analyses with it.

My lab and I are genuinely considering publishing our results that came about from using this new tool that I just installed for the purposes of conducting this review, so consider that a vote of confidence for the approach.

Overall the methods are easy to follow, the installation and use was similarly easy, the manuscript was well-written, and the statistical approaches are sound (with the caveat, as I mentioned earlier, that I am not an imaging expert).

Of course I have a few comments and suggestions for the authors, which I will itemize below.

*1) First, I think the authors should emphasize that this data integration approach is a *beginning*, not an end. They discuss this, but I think they underemphasize it. Now that this integration framework exists, it should in theory only become more useful and more powerful as more data and data types are pulled into it.*

We agree with the Reviewer that the evolving data resource is a key strength of neuromaps, and thank the Reviewer for the encouraging comments. This comment is similar in spirit to the Reviewer's next comment (Comment #2) about emphasizing the strength of the contextualizing factor. We address both comments together.

We emphasize that the data integration approach is the start to an expanding resource, and that with this expansion comes more interdisciplinary research ("Results" section, "The neuromaps data repository" subsection, Paragraph #3):

"The neuromaps toolbox makes it easy to contextualize novel brain maps to a range of molecular, structural, temporal, and functional features. This will facilitate an expansion of research questions, allowing researchers to bridge brain topographies across several spatial scales and across disciplines outside of their immediate scope (Voytek & Voytek, 2012). Importantly, the included brain maps are only the start of the neuromaps data repository. The contribution pipeline lets researchers add vertex/voxel-level brain maps to the toolbox, pending approval from the maintainers. The data repository will therefore become an increasingly rich resource of structural and functional brain annotations. As the toolbox expands, it will become more comprehensive, giving the neuroscience field the power to identify cross-disciplinary associations. Information on contributing new brain maps can be found in the online documentation for the software (<https://netneurolab.github.io/neuromaps/>)."

We also devote a new paragraph in the discussion to the future potential of neuromaps (“Discussion” section, Paragraph #2):

“Modern technological and data sharing advances have increasingly moved neuroscience research towards integrative questions rooted in data science. Imaging, recording, tracing and sequencing technology offer an ability to quantify multiple features of neuroanatomy and function with unprecedented detail and depth. How to bring these disparate data types into an accessible analytic framework has remained a challenge.

The neuromaps software package offers flexible tools to the neuroscience community for transforming and parcellating data, as well as making statistically robust enrichment analysis of brain maps. In particular, neuromaps provides an ever-expanding integrative atlas of structural and functional annotations, including genes, receptors and cell types. Furthermore, as neuromaps is adopted by the community, novel annotations from emerging technologies and datasets can be added by users, resulting in an organic and perpetually evolving resource. This will allow new maps to be systematically contextualized with respect to multiple canonical annotations from diverse data types and disciplines, resulting in (1) standardized reporting of new results, and (2) inspiration for mechanistic follow-ups. Ultimately, neuromaps is a first step towards integrative analytics for multi-modal, multi-scale neuroscience.”

2) The authors also undersell, in my opinion, the strength of the contextualization factor that neuromaps permits: currently the vast majority of labs collect data in a vacuum, focusing on their primary approach. However neuromaps provides an easy framework for placing results in the broader context of what is known about the spatial patterns of effects as relate to cognition and disease (NeuroSynth), receptor densities (PET), gene expression (Allen Institute), etc.

Please see our response to the previous comment (Comment #1).

3) Figure 2, column 3 is labeled "electrophysiology", but technically this is MEG, correct? So it's more "magneto physiology".

The data is indeed MEG data but we prefer to use the term “electrophysiology” for two reasons:

1. The headings in Figure 2 are intended to be broad categories of brain annotations. In the case of “electrophysiology”, this category will hopefully welcome the addition of other electrophysiological data i.e. EEG or ECoG data.
2. This label is in line with recent efforts to position MEG data within the broader electrophysiological field (Baillet, 2017).

Baillet, S. (2017). Magnetoencephalography for brain electrophysiology and imaging. *Nature neuroscience*, 20(3), 327-339.

4) For the gene expression data, my understanding is that expression is sampled from points. The authors state that they don't up-sample their data--rather they try to keep in in the native format--but I can't figure out from the methods how these point samples of gene expression were used to generate surface maps.

We thank the Reviewer for catching this caveat as gene expression maps were indeed upsampled using a KNN interpolation. We now address this directly in the text:
(Figure 2 caption):

“Note that: (1) two of the maps (second column: evo and devel expansion) only have data for the right hemisphere, (2) the intrinsic timescale is log-transformed, (3) a selected four of the 36 neurotransmitter receptor maps are shown here (Hansen et al., 2021, Kantonen et al., 2020, Naganawa et al., 2021, Savli et al., 2012, Smart et al., 2019), and (4) the genomic gradient is upsampled to the fsaverage 10k surface (for accessing raw and processed AHBA data, see <https://abagen.readthedocs.io/en/stable/> (Markello et al., 2021)).”

(“Results” section, “The neuromaps data repository” subsection, Paragraph #2):

“All brain maps except for the AHBA-derived genomic gradient are provided in the original coordinate system in which they were defined to avoid errors caused by successive interpolation. The AHBA genomic gradient was upsampled to the fsaverage 10k surface using a KNN interpolation. Raw and processed AHBA data can be accessed at <https://abagen.readthedocs.io/en/stable/> (Markello et al., 2021).”

(“Methods” section , “Published brain maps” subsection):

“Maps were obtained in their original coordinate system, with the exception of the AHBA-derived genomic gradient. AHBA samples across the surface were upsampled to the fsaverage 10k surface using a KNN interpolation before applying principal component analysis.”

5) For Figure 5b, the embedded network image is interesting, but without labels for each of the maps, it's difficult to understand what the takeaway is.

We have reworked this panel to include labels on the brain surfaces. This figure is intended to highlight the observation that many well-studied “gradients” in the broader literature tend to fall within two clusters: one that follows an anterior-posterior gradient, and one that follows a unimodal-transmodal

gradient. Note that the complete figure alongside any new text is included in our response to the next comment.

6) For Figure 5c, I get the point that arbitrary p-value thresholds can give misleading results, especially considering the flexibility of analysis methods, but these are *very* small differences. Can the authors quantify the distribution of these differences between coordinate systems, perhaps showing a histogram of all the delta-r values for all pairs?

We thank the Reviewer for this suggestion and we have now added a new panel to Figure 5 showing how much correlation coefficients change across coordinate systems. Specifically, for a specific pair of annotations in a specific coordinate system, we compare their correlation to the correlation calculated in the five other spaces/resolutions. We plot the distribution of change in correlation (delta r) for each coordinate system. In other words, each histogram represents the 5 delta r's across every pair of annotations. This not only shows that correlation coefficients change minimally, but also how coordinate systems compare to one another in terms of how different they are.

Figure 5:

a | brain map correlations

b | transformation-induced correlation differences

c | map associations

d | embedded network (fsLR 32k)

(“Results” section, “Demonstrating neuromaps” subsection, Paragraph #4):

“Figure 5a shows that for most map-to-map comparisons, choice of coordinate system has a minimal effect: correlation coefficients on average by only change by $|r|=0.018$ (variance of absolute difference = 0.0002, range = [-0.10, 0.079]). Figure 5b shows the distribution of correlation changes (delta r), comparing correlations between brain annotations in different coordinate spaces. Specifically, for a given pair of brain annotations, we compute their correlation in the six available surface spaces/resolutions (listed in Figure 5a). Next, we plot the distribution of correlation differences between each space and a “constant” space. Each histogram represents the distribution when a different coordinate space/resolution is defined as the “constant” space. Although the changes are minimal, there are instances in which associations between maps are statistically significant in one coordinate system and not significant in another (Figure 5c); however, in most cases the p-value for these

relationships was close to the statistical alpha (i.e., $p < 0.05$) such that the actual effect size only changed by $r \leq 0.10$. Across all examined systems, we find that the brain maps tend to form two distinct clusters (Figure 5d), largely recapitulating previously-established relationships observing anterior-posterior and unimodal-transmodal axes of variation (Margulies et al., 2016, Hansen et al., 2021, Shafiei et al., 2020). These results are encouraging, suggesting that transforming brain annotations between different systems generally preserves their relationships.”

7) Relatedly, the authors state that, "choice of coordinate system has a minimal effect: correlations between maps on average only change by $|r| = 0.018$." However without understanding the range, variance, mean, etc. it's impossible to know if this a "big" or "small" effect.

We now report the mean, variance, and range of delta r (change in correlation) across all six histograms (please see our response to the comment above).

(“Results” section, “Demonstrating neuromaps” subsection, Paragraph #4):

“Figure 5a shows that for most map-to-map comparisons, choice of coordinate system has a minimal effect: correlation coefficients on average by only change by $|r|=0.018$ (variance of absolute difference = 0.0002, range = [-0.10, 0.079]).”

8) There are a few results where no statistical effect sizes are given. Specifically the statement that, "schizophrenia-related cortical thinning is enriched in areas that experience the greatest neurodevelopmental expansion," and, "we find no evidence to support recent claims that schizophrenia targets specific parts of the unimodal- transmodal processing hierarchy."

We now include effect size and p-value in the text:

(“Results” section, “Demonstrating the neuromaps toolbox” subsection, Paragraph #1):

“We find that schizophrenia-related cortical thinning is enriched in areas that experience the greatest neurodevelopmental expansion ($r_{NIH}=0.26$, $p_{spin}=0.001$; $r_{PNC}=0.29$, $p_{spin}=0.001$), consistent with the notion that schizophrenia is a neurodevelopmental disease that affects adolescent development (Murray et al., 1987, Rapoport et al., 2012). On the other hand, we find no evidence to support recent claims that schizophrenia targets specific parts of the unimodal-transmodal processing hierarchy ($r_{FC}=-0.08$, $p_{spin}=0.35$; Dong et al., 2021, Dong et al., 2020).”

(“Results” section, “Demonstrating the neuromaps toolbox” subsection, Paragraph #2):

“Contextualizing this source map with respect to the other 12 selected target maps from the toolbox, we find that areas with the greatest evolutionary expansion have the greatest inter-individual variability of regional functional connectivity profiles ($r=0.58$, $p_{spin}=0.005$; Mueller et al., 2013). This is consistent with the notion that the greatest inter-individual differences in brain structure and function are in phylogenetically newer transmodal cortex (Sydnor et al., 2021). Moreover, we find significant anticorrelations with the genomic PC1 ($r=-0.51$, $p_{spin}=0.027$) and intracortical myelin ($r=-0.46$, $p_{spin}=0.005$), consistent with the idea that phylogenetically newer cortex is characterized by divergence from genomic gradients and canonical stimulus-response circuit configurations (Buckner & Krienen, 2013). Interestingly, we also find that the map is anticorrelated with cerebral blood volume - a novel result that suggests that phylogenetic adaptation is concomitant with greater metabolic efficiency ($r=-0.37$, $p_{spin}=0.003$).”

*9) There are a lot of required packages for neuromaps, and some of them seem like "nice to have" packages, but I don't know why they're in the requirements.txt. For example, why are IPython and Jupiter *required*? Similarly, Matplotlib and Seaborn (and possibly SurfPlot) are nice to have for plotting, but aren't fundamentally required for any of the actual computational processing or data analysis.*

We have limited the neuromaps requirement list to seven packages (scipy, scikit-learn, numpy, Nilearn, nibabel, matplotlib, packaging). We keep matplotlib for the sake of the plotting functionalities in neuromaps. The revised requirement list can be found here:

<https://github.com/netneurolab/neuromaps/blob/main/setup.cfg#L25-L31> and here:

<https://github.com/netneurolab/neuromaps/blob/main/requirements.txt>.

Decision Letter, first revision:

Dear Dr. Misić,

Thank you for submitting your revised manuscript "Neuromaps: structural and functional interpretation of brain maps" (NMEMH-A48201A). It has now been seen by the original referees and their comments are below. The reviewers find that the paper has improved in revision, and therefore we'll be happy in principle to publish it in Nature Methods, pending minor revisions to satisfy the referees' final requests and to comply with our editorial and formatting guidelines.

TRANSPARENT PEER REVIEW

Nature Methods offers a transparent peer review option for new original research manuscripts submitted from 17th February 2021. We encourage increased transparency in peer review by publishing the reviewer comments, author rebuttal letters and editorial decision letters if the authors agree. Such peer review material is made available as a supplementary peer review file. Please state in the cover letter 'I wish to participate in transparent peer review' if you want to opt in, or 'I do not wish to participate in transparent peer review' if you don't. Failure to state your preference will result in delays in accepting your manuscript for publication.

Thank you again for your interest in Nature Methods Please do not hesitate to contact me if you have any questions.

Best regards,
Nina

Nina Vogt, PhD
Senior Editor
Nature Methods

ORCID

Reviewer #1 (Remarks to the Author):

I thank the authors for their reply. My comments have been fully addressed.

Reviewer #2 (Remarks to the Author):

The authors did an excellent job addressing all my comments. They strengthened the paper, figures and the software. I am excited to see this paper published and for this tool to be used in neuroimaging communities.

Reviewer #3 (Remarks to the Author):

The authors have done a great job responding to all of my comments and questions. Thank you for building such a great new resource!

Final Decision Letter:

Dear Bratislav,

I am pleased to inform you that your Article, "Neuromaps: structural and functional interpretation of brain maps", has now been accepted for publication in Nature Methods. Your paper is tentatively scheduled for publication in our October print issue, and will be published online prior to that. The received and accepted dates will be January 25th, 2022 and August 24th, 2022. This note is intended to let you know what to expect from us over the next month or so, and to let you know where to address any further questions.

Please note that Nature Methods is a Transformative Journal (TJ). Authors may publish their research with us through the traditional subscription access route or make their paper immediately open access through payment of an article-processing charge (APC). Authors will not be required to make a final decision about access to their article until it has been accepted. Find out more about Transformative Journals

Authors may need to take specific actions to achieve compliance with funder and institutional open access mandates. If your research is supported by a funder that requires immediate open access (e.g. according to Plan S principles) then you should select the gold OA route, and we will direct you to the

compliant route where possible. For authors selecting the subscription publication route, the journal's standard licensing terms will need to be accepted, including self-archiving policies. Those licensing terms will supersede any other terms that the author or any third party may assert apply to any version of the manuscript.

Your paper will now be copyedited to ensure that it conforms to Nature Methods style. Once proofs are generated, they will be sent to you electronically and you will be asked to send a corrected version within 24 hours. It is extremely important that you let us know now whether you will be difficult to contact over the next month. If this is the case, we ask that you send us the contact information (email, phone and fax) of someone who will be able to check the proofs and deal with any last-minute problems.

If, when you receive your proof, you cannot meet the deadline, please inform us at rjsproduction@springernature.com immediately.

Once your manuscript is typeset and you have completed the appropriate grant of rights, you will receive a link to your electronic proof via email with a request to make any corrections within 48 hours. If, when you receive your proof, you cannot meet this deadline, please inform us at rjsproduction@springernature.com immediately.

Once your paper has been scheduled for online publication, the Nature press office will be in touch to confirm the details.

Content is published online weekly on Mondays and Thursdays, and the embargo is set at 16:00 London time (GMT)/11:00 am US Eastern time (EST) on the day of publication. If you need to know the exact publication date or when the news embargo will be lifted, please contact our press office after you have submitted your proof corrections. Now is the time to inform your Public Relations or Press Office about your paper, as they might be interested in promoting its publication. This will allow them time to prepare an accurate and satisfactory press release. Include your manuscript tracking number NMETH-A48201B and the name of the journal, which they will need when they contact our office.

About one week before your paper is published online, we shall be distributing a press release to news organizations worldwide, which may include details of your work. We are happy for your institution or funding agency to prepare its own press release, but it must mention the embargo date and Nature Methods. Our Press Office will contact you closer to the time of publication, but if you or your Press Office have any inquiries in the meantime, please contact press@nature.com.

Nature Portfolio journals encourage authors to share their step-by-step experimental protocols on a protocol sharing platform of their choice. Nature Portfolio 's Protocol Exchange is a free-to-use and open resource for protocols; protocols deposited in Protocol Exchange are citable and can be linked from the published article. More details can found at www.nature.com/protocolexchange/about.

Please note that you and any of your coauthors will be able to order reprints and single copies of the issue containing your article through Nature Portfolio 's reprint website, which is located at <http://www.nature.com/reprints/author-reprints.html>. If there are any questions about reprints please send an email to author-reprints@nature.com and someone will assist you.

Best regards,
Nina

Nina Vogt, PhD
Senior Editor
Nature Methods